# Internal control beliefs shape positive affect and associated neural dynamics during outcome valuation

David S. Stolz 🆔 [1✉], Laura Müller-Pinzler[1], Sören Krach[1] & Frieder M. Paulus 🆔 [1✉]

Experiencing events as controllable is essential for human well-being. Based on classic psychological theory, we test how internal control beliefs impact the affective valuation of task outcomes, neural dynamics and ensuing behavioral preferences. In three consecutive studies we show that dynamics in positive affect increase, with a qualitative shift towards self-evaluative pride, when agents believe they caused a given outcome. We demonstrate that these outcomes engage brain networks processing self-referential information in the cortical midline. Here, activity in the ventromedial prefrontal cortex tracks outcome valence regarding both success as well as internal control, and covaries with positive affect in response to outcomes. These affective dynamics also relate to increased functional coupling between the ventral striatum and cortical midline structures. Finally, we show that pride predicts preferences for control, even at monetary costs. Our investigations extend recent models of positive affect and well-being, and emphasize that control beliefs drive intrinsic motivation.

[1] Social Neuroscience Lab at the Translational Psychiatry Unit (TPU), Department of Psychiatry and Psychotherapy, University of Lübeck, Ratzeburger Allee 160, D-23538 Lübeck, Germany. ✉email: david.stolz@uni-luebeck.de; frieder.paulus@uni-luebeck.de

The subjective belief of being in control over events in one's life is essential for well-being[1–3]. The ways in which individuals perceive themselves or external forces as determining their fate have been conceptualized in the theory of locus of control[4], and since then have deeply influenced theory formation in psychology[5–7]. Here, the sense of being in control hinges on the subjective belief that the course of events can be shaped by own efforts and actions (i.e., internal control), creating the cognitive foundation of attributing their outcomes to the self. Internal control beliefs can determine whether individuals will show effort[8], make career choices[9], and are also more generally considered to be a protective factor for various psychiatric phenomena[10,11]. In contrast, the conviction of having no control and thus being at the mercy of chance or other forces in one's environment (i.e., external control) has been linked to learned helplessness and depression[12].

Having opportunities to choose is a fundamental prerequisite of exerting control[3]. Choice opportunities allow to pick between options with different value[13], and thus are a means for maximizing the probability of achieving desired outcomes and reducing uncertainty[14]. The importance of exerting control over one's environment has stimulated the idea that having choices per se carries intrinsic value[3]. Studies have demonstrated that humans favor choice options that are followed by a second choice over those that are not[15], prefer tasks with more over those with fewer choice options[16], and value the opportunity to choose in a gamble[17–19]. In this line, cues signaling an upcoming choice were associated with increased activity in the ventral striatum (VS)[18,19], a brain region implicated in dopaminergic reward processing[20–22].

Although having choices is the condition precedent for exerting control, the lynchpin of psychological theory relating control beliefs to well-being are self-related thoughts and subjective models of whether an outcome can be achieved due to the capabilities of the agent[1,2,4,6]. Thereby, the experience of control should increase if internal models of behavior are available that explain how actions—if executed correctly—yield a specific outcome (as e.g., when playing darts)[23], in contrast to settings in which behavior is thought to be linked to its outcomes by chance alone (e.g., when playing roulette)[18,19]. Thus, while having choices is the condition necessary for exerting control, it is in principle not sufficient for experiencing the quality of control belief underlying self-attribution of task outcomes[4,6,7]. Only if the context offers the potential for building the belief of internal control over outcomes, it is possible to attribute events to the self, own efforts, and abilities, with broader implications for self-related affect and motivation[24,25].

Theories on self-conscious affect[26] and appraisal theories[27,28] predict that control beliefs shift the valuation of outcomes and shape affective experiences. While internal control beliefs should relate to dynamics in positive affect due to the added value, the attribution of outcomes to internal causes is also a necessary condition for the experience of the self-conscious affect of pride[24]. Pride essentially hinges on a subjective model of control—the belief that an outcome was caused by one's own actions—resulting in self-approval in case the event is relevant for personal goals[27,29,30]. Pride experiences thus underlie self-esteem[31], foster intrinsic motivation[25], and mediate the contribution of internal control beliefs to well-being[2,31]. In contrast, positive events caused by uncontrollable factors such as winning a game of roulette also drive affective dynamics such as momentary happiness[32,33], but might not have similar consequences in terms of self-evaluation and motivation[25]. In this line, earnings that have been obtained through own efforts and labor are valued more than windfall gains[34], thereby contributing to the effort paradox that the value of effort sometimes exceeds its costs[35].

In this study, we test how the affective valuation of outcomes varies according to how much a task allows for developing internal control beliefs and self-attribution, how this relates to shifts in neural processing of objectively similar task outcomes, and provide evidence that the subjective value of internal control relates to dynamics of self-evaluative affect. As predicted by theory, positive affect and pride in particular should be more strongly modulated by outcomes that are perceived to depend on one's own performance[24,29]. Aside from the ventral striatum, prior evidence suggests the ventromedial prefrontal cortex (vmPFC) to be a key cortical interface for integrating self-attribution during outcome valuation. The vmPFC has extensive connections with the ventral tegmental area, the amygdala and the striatum[36], is associated with domain general computations of value[13,37], and has been linked to behavioral control and persistence in the face of failure[38–40]. Further, the vmPFC belongs to structures of the cortical midline implicated in self-related processing (cortical midline structures; CMS)[41,42]. Specifically, the vmPFC has been associated with the generation of affective meaning[43], the value of revealing information about the self[44], and updating of self-esteem and self-relevant optimistic beliefs[45,46]. Furthermore, vmPFC activity is correlated with positive social evaluative feedback[47,48], and predicts increased self-enhancement under conditions of self-evaluative threat[48]. In addition, the vmPFC encodes representational information from which both the self-relevance and the positivity of stimuli can be decoded in separate tasks[49].

In order to manipulate internal control beliefs and investigate their impact on affect, neural dynamics, and motivation, we conducted three experiments ($N = 129$) in which we stimulate control beliefs beyond having a choice or not, inducing low, medium, and high levels of perceived internal control (LC, MC, and HC). In study 1, we show that affective responses to outcomes depend on internal control beliefs as predicted by theory. In study 2, we characterize the affective and neural dynamics in response to outcomes obtained at different levels of internal control. We demonstrate that brain regions implicated in processing reward value and self-reference, specifically the vmPFC, are linked to internal control beliefs and affect. As a last step, in study 3 we predict the subjective value of controllable environments based on the dynamics of self-related positive affect. Our results support the broad impact of internal control beliefs on affect, motivated behavior, and underlying brain function.

## Results

**Internal control and affective dynamics**. Study 1 manipulated levels of internal control beliefs above and beyond having a choice or not. In brief, we presented participants ($N = 40$; see Supplementary Table 1) with a grid of one centered gray square and 32 surrounding squares that were separated into two parts of shaded colors (see Fig. 1). On the lowest level of control, successes (WIN outcomes; 0.20 €) and non-successes (noWIN outcomes; 0.00 €) depended on an automated gamble, which was initiated by clicking on the gray square (low control, LC). Comparable with previous studies[18,19], in the LC condition participants thus had no option to choose. On the medium level of control, WIN and noWIN outcomes allegedly depended on the correct choice between the two colors (medium control, MC) introducing the opportunity to choose in MC. On the highest level of control, WIN and noWIN outcomes allegedly depended on whether subjects were able to identify the brightest square within the shades of one color (high control, HC) thus allowing the attribution of task outcomes to internal causes. Unknown to the participants, across all three conditions outcome probabilities were held constant at 50% by predefined feedback (see "Methods"

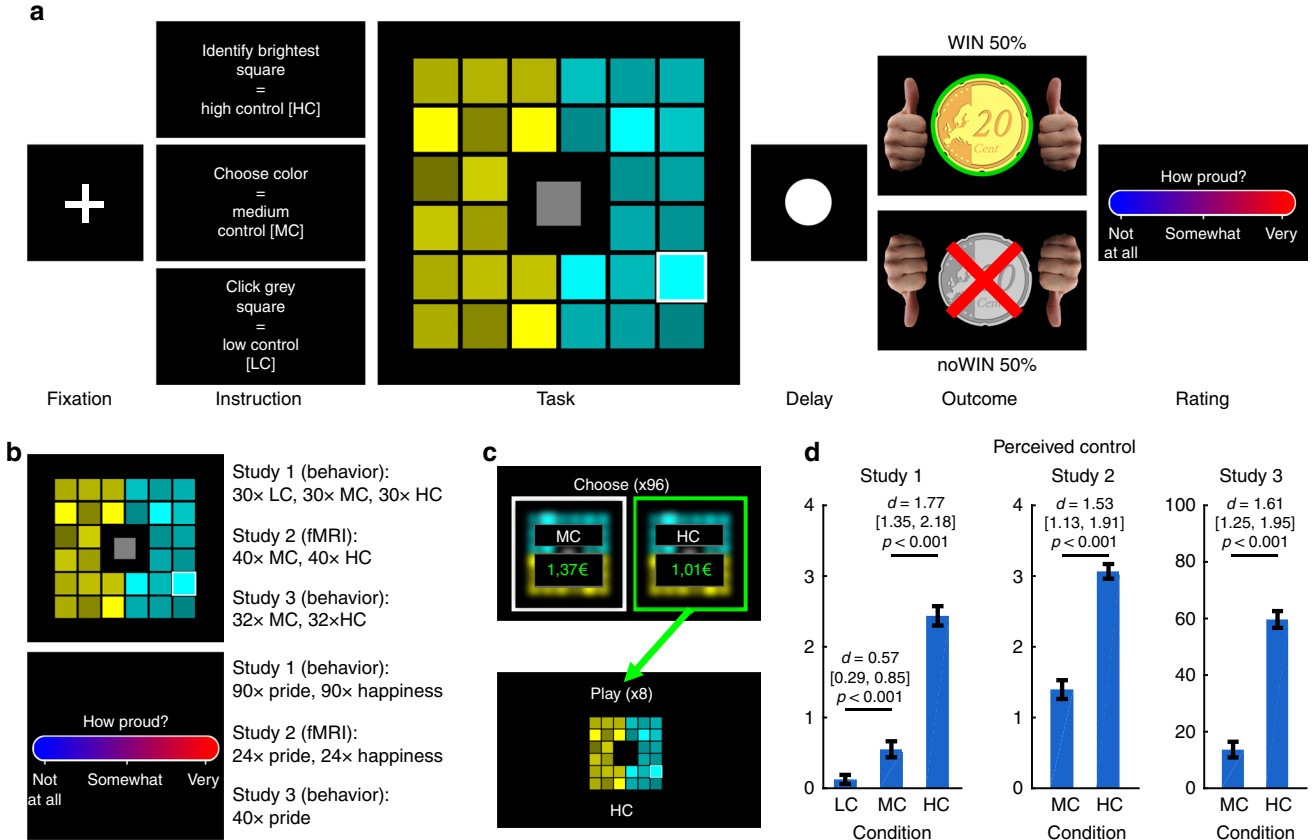

**Fig. 1 Overview about the experimental paradigm and the three studies. a** Trial structure of the task. A fixation cross was followed by a cue indicating that the participant has to find the brightest square within either turquoise or yellow (high control, HC), choose between yellow and turquoise (medium control, MC), or play a gamble by clicking on the gray square in the middle (low control, LC; only presented in study 1). After the cue, the task grid occurred, and was followed by a delay and task outcomes. Pride and happiness ratings regarding the preceding outcome were presented after the outcome phase. See "Methods" and Supplementary Methods for detailed timing information of the three studies. **b** Overview of the three studies. The task was presented in each study with varying numbers of trials, conditions, and ratings. In study 1, both pride and happiness were rated after each outcome. In study 2, either pride or happiness were rated after a given outcome. Ratings were evenly distributed across the four possible outcomes, and the order of pride and happiness ratings was predefined and counterbalanced across subjects. In study 3, only pride was rated, and predefined ratings were evenly distributed across all possible outcomes. See "Methods" for details. **c** In study 3, after the main task, 96 trials of a choice task were presented. The choice options offered to play either HC or MC with adaptively changing amounts of money determined using a staircase algorithm. Participants were informed that some of the chosen options would be actually presented, and that the associated monetary gain would be awarded at the end of the experiment. **d** Internal control belief ratings for each study ("How much could you influence the outcome of the task"). d = Cohen's d. Values in brackets represent lower and upper bounds of 90% confidence intervals. HC high control, MC medium control, LC low control. Error bars are +/− 1 standard error of the mean. Source data are provided as a Source Data file.

for details). This is a key step since with increasing internal control beliefs, outcomes also get more predictable and less uncertain in everyday life[7,50], and we did not want to capture effects related to the mere probability of achieving an outcome.

Subjects experienced the HC condition as more controllable than the other conditions (main effect of condition: $F(1.74, 76.72) = 133.97$, $p < 0.001$; planned comparisons: HC > MC: $t(39) = 11.21$, $p < 0.001$, $d = 1.77$, 90% CI = [1.35; 2.18]; MC > LC: $t(39) = 3.60$, $p < 0.001$ $d = 0.57$, 90% CI = [0.29; 0.85]; Bonferroni-corrected for multiple comparisons, one-sided). This replicates previous findings showing that choice is essential for the experience of control[18,19], but additionally shows that perceived control comparably excels in situations in which only one's own performance and abilities allow to obtain the desired outcomes. As predicted by theory, the dynamics in the pride experiences induced by succeeding (WIN vs noWIN) depended more strongly on the task than it was the case for happiness ratings (condition*affect*outcome valence: $F(1.29, 50.10) = 11.44$,

$p < 0.001$, $\eta_p^2 = 0.23$; Fig. 2, for full rmANOVA effects, see Supplementary Table 2). Affect ratings were higher for WIN than for noWIN outcomes for both happiness (LC: $t(39)=7.85$, one-sided $p < 0.001$; MC: $t(39) = 8.16$, $p < 0.001$; HC: $t(39) = 9.74$, $p < 0.001$) and pride (LC: $t(39) = 3.29$, $p < 0.001$; MC: $t(39) = 4.41$, $p < 0.001$; HC: $t(39) = 7.37$, $p < 0.001$, corrected). These differences were larger in HC than in MC (pride: $t(39) = 6.97$, one-sided $p < 0.001$; happiness: $t(39) = 4.64$, one-sided $p < 0.001$), and larger in MC than in LC for pride ($t(39) = 3.03$, one-sided $p = 0.013$), and for happiness ($t(39) = 2.24$, one-sided $p = 0.047$, corrected). To test whether internal control over successes impacts pride more strongly than happiness, we compared the difference between HC:WIN and MC:WIN outcomes specifically. This increase was stronger for pride than for happiness ratings ($t(39) = 4.039$, one-sided $p < 0.001$), while the overall intensity of happiness was higher in all four conditions (see Fig. 2).

The main effects of outcome valence ($F(1.00, 39.00) = 60.92$, $p < 0.001$, $\eta_p^2 = 0.61$) and condition ($F(1.31, 50.93) = 60.92$,

                    3

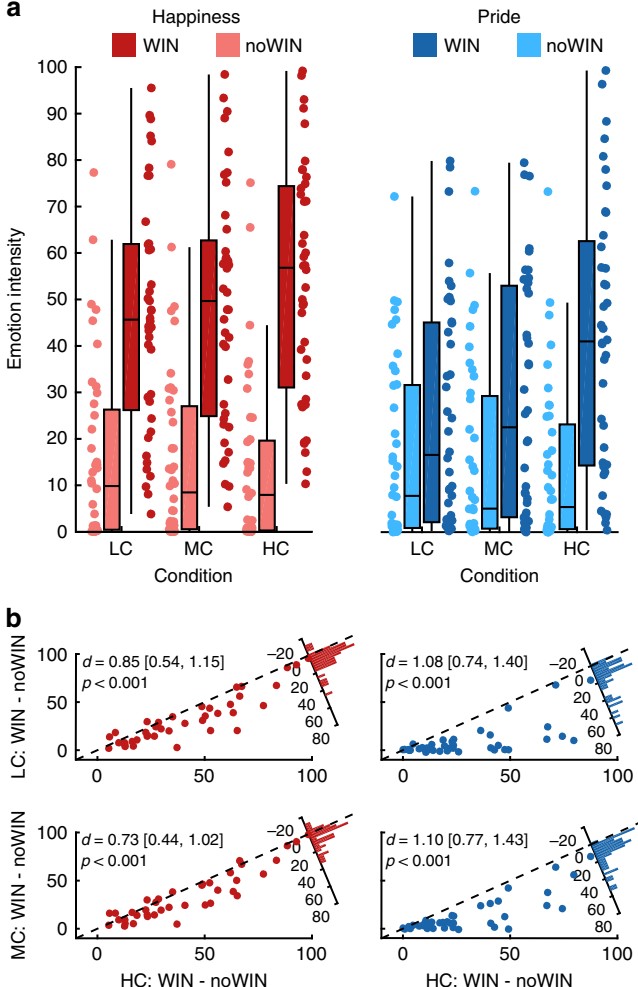

**Fig. 2 Task effects on emotion ratings. a** Emotion ratings from study 1, separate for pride and happiness, high, medium, and low control as well as outcome valence. **b** Scatterplots depict reactivity to WIN vs noWIN outcomes for happiness (left) and pride (right) in dependence of perceived control (x-axis: HC; y-axis (top): LC; y-axis (bottom): MC). Top and bottom rows depict relationships between emotion reactivity in LC and HC, as well as MC and HC, respectively. Histograms in top right corners visualize the differences in emotion reactivity between HC and LC (top row), or HC and MC (bottom row). HC high control, MC medium control, LC low control. d = Cohen's d. Values in brackets represent 90% confidence intervals for d. The center line of boxplots depicts the median, the upper and lower box borders are the 25th and 75th percentile, and the length of the whiskers is 1.5 times the interquartile range. Source data are provided as a Source Data file.

$p < 0.001$, $\eta_p^2 = 0.43$), as well as their two-way interaction ($F(1.37, 53.55) = 48.51$, $p < 0.001$, $\eta_p^2 = 0.55$) additionally show how internal control beliefs drive positive affect in general in response to stimulus outcomes. Regardless of the strong association between pride and happiness ratings (within-subject Pearson's r: $M = 0.67$, SD = 0.21, $t(39) = 20.29$, one-sided $p < 0.001$; for full report of between-subject correlations see Supplementary Tables 11 and 13), the pride response (i.e., the difference [HC: WIN–HC:noWIN]–[MC:WIN–MC:noWIN]) increased more strongly from MC to HC than the happiness response ($t(39) = 3.69$, one-sided $p < 0.001$). No difference was found when comparing pride and happiness responses between LC and MC ($t(39) = 1.57$, one-sided $p = 0.124$, corrected). These findings

demonstrate that internal control beliefs depend on contexts which allow the formation of subjective models that explain how outcomes can be achieved through own actions and mastering the task. The implied self-relevance of the outcomes changes the affective valuation by driving positive affect in general, and by qualitatively shifting the construal of the experience toward pride[24]. Having a choice may be rewarding in itself, but falls short of internal control beliefs as defined in classic theories[4,6,7] with less impact on this self-evaluative aspects of positive affect[26].

**Neural markers of outcome valuation under internal control.** Using fMRI ($N = 39$), we replicated and extended our findings from study 1 by characterizing the neural processing of outcomes achieved under internal control and their impact on affective dynamics. Performing the HC and MC conditions (LC was not included, see "Methods") subjects again experienced higher internal control in HC than in MC (paired t test: $t(38) = 9.56$, one-sided $p < 0.001$, $d = 1.53$, 90% CI = [1.13; 1.91]). Affect ratings differed between tasks according to the level of control and outcome valence, similar to study 1 (condition*outcome valence*affect: $F(1, 38) = 9.08$, $p = 0.005$, $\eta_p^2 = 0.19$; Fig. 3; for full rmANOVA effects, see Supplementary Table 3), with the pride response being significantly larger than the happiness response ($t(38) = 3.01$, one-sided $p = 0.005$). This emphasizes how the construal of self-related positive affect hinges on the subjective belief that outcomes depend on one's own contributions. Underlining the effect of internal control on affective responses, the difference in pride between HC:WIN and MC:WIN outcomes was significantly larger than the equivalent term for happiness ($t(38) = 2.82$, one-sided $p = 0.004$).

Receiving a positive outcome (WIN > noWIN) yielded significant activations in regions associated with processing of value (as indicated by ROI-analyses using the VALUE¬SELF mask, covering areas associated with the term value, but not self-referential; created using Neurosynth[51]; see Supplementary Methods). Specifically, we observed significant activation with peaks in the bilateral VS (left: x, y, z (mm): −14, 8, −10, $t(114) = 8.18$; $Z = 7.24$, $k = 235$; right: 16, 4, −12, $t(114) = 7.69$; $Z = 6.89$, $k = 233$; all coordinates in MNI space), anterior cingulate cortex (ACC; −2, 44, −6, $t(114) = 6.71$; $Z = 6.15$, $k = 108$), superior medial frontal gyrus (−2, 58, 0, $t(114) = 8.14$; $Z = 7.21$, $k = 61$), and vmPFC (4, 48, −2, $t(114) = 5.59$; $Z = 5.24$, $k = 12$; all $p$-values < 0.05, FWE-corrected; for whole-brain effects see Supplementary Tables 14–16).

Receiving outcomes attributable to one's own actions (HC > MC) yielded significant activations in CMS (ROI analysis using the SELF¬VALUE mask, covering regions associated with the term self-referential, but not value), with peaks in left vmPFC (−8, 50, −12, $t(114) = 6.34$; $Z = 5.85$, $k = 27$), and left ACC (−2, 38, 4, $t(114) = 6.27$; $Z = 5.80$, $k = 108$). In addition, this contrast yielded significant effects in posterior CMS, specifically the left cuneus (−8, −58, 20, $t(114) = 6.61$; $Z = 6.07$, $k = 208$), extending to precuneus (−6, −50, 6, $t(114) = 5.60$; $Z = 5.25$, $k = 14$; all $p$-values < 0.05, FWE-corrected). Furthermore, we found significant activations in aspects of the vmPFC covered by the SELF ∩ VALUE mask (covering regions associated with both self-referential and value, −8, 48, −12, $t(114) = 6.13$, $Z = 5.69$, k = 164) as well as the VALUE¬SELF mask (−6, 52, −12, $t(114) = 5.71$, $Z = 5.35$, $k = 133$; all $p$-values < 0.05, FWE-corrected; see Supplementary Tables 17–19). These findings indicate that cortical regions associated with processing of self-relevant information and value are engaged when outcomes are obtained in task environments that are perceived as controllable and thus provide information about one's capabilities[30,52].

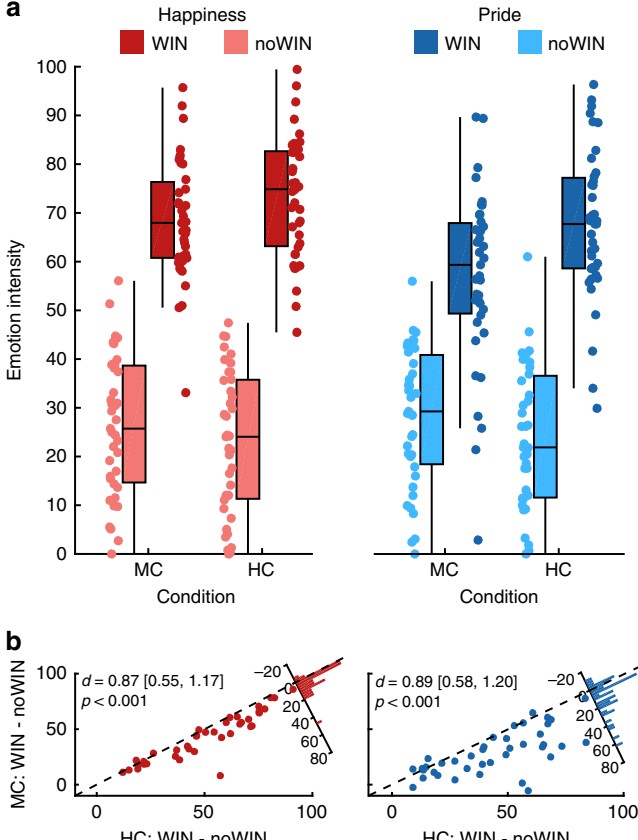

**Fig. 3 Task effects on emotion ratings in study 2. a** Emotion ratings, separate for pride and happiness, high and medium control as well as outcome valence. **b** Scatterplots depict reactivity to WIN vs noWIN outcomes for happiness (left) and pride (right) in dependence of internal control condition (x-axis: HC; y-axis: MC). Histograms in top right corners visualize the differences in emotion reactivity between HC and MC. HC high control, MC medium control. d = Cohen's d. Values in brackets represent 90% confidence intervals for d. The center line of boxplots depicts the median, the upper and lower box borders are the 25th and 75th percentile, and the length of the whiskers is 1.5 times the interquartile range. Source data are provided as a Source Data file.

Next, a two-way conjunction contrasting wins achieved under high control to both, wins achieved under medium control and no wins achieved under high control ([HC:WIN > MC:WIN] ∩ [HC:WIN > HC:noWIN]; Fig. 5) revealed the joint effects of winning and high control in the left vmPFC when subjects' successes were attributable to own actions (−4, 48, −10, t(114) = 3.45; Z = 3.36, k = 12, p = 0.022, FWE-corrected within the SELF ∩ VALUE mask; see Supplementary Tables 20–21). This more direct test for the additivity of the effects in this region, compared with the overlapping response to both winning and control (see Fig. 4), reflects how the processing of value and self-reference converges in the vmPFC[46,53]. No super-additive effects were found in the sense of a significant interaction even at more lenient thresholds at p < 0.001 uncorrected within the set of a priori defined regions of interests.

Prior studies have demonstrated that the anticipation to exert control increases activation in the ventral striatum[18,19]. We therefore tested for differential VS activity during the presentation of the task cues, contrasting HC against MC. This analysis yielded significant activations in the bilateral VS (left: −12, 8, −6, t(38) = 4.73; Z = 4.16, k = 26, p = 0.013; right: 18, 4, −6, t(38) = 4.85, Z = 4.25, k = 62, p = 0.009; FWE-corrected inside the

VALUE¬SELF mask, see Supplementary Table 22 and Supplementary Fig. 1), replicating previous findings of increased striatal activation when environmental information signals the potential for exerting control[18,19].

Within-subject dynamics in happiness were significantly correlated with neural activity within the VALUE¬SELF mask at outcome presentation (16, 2, −12, t(38) = 4.55, Z = 4.04, k = 3, p = 0.022; see Fig. 5), as well as the SELF ∩ VALUE mask (left vmPFC; −2, 48, −4, t(38) = 4.60, Z = 4.08, k = 47, p = .002). Within-subject variability in pride ratings showed more extended effects in the SELF¬VALUE mask (dorsal medial prefrontal cortex (dmPFC); −2, 56, 2, t(38) = 5.96, Z = 4.98, k = 122, p = 0.001) and the VALUE¬SELF mask (dmPFC; −2, 56, 0, t(38) = 6.15, Z = 5.09, k = 22, p < 0.001). In addition, variability in pride ratings correlated significantly with BOLD responses in the SELF ∩ VALUE mask (left vmPFC; −4, 48, −6, t(38) = 4.83, Z = 4.24, k = 94, p = 0.001; all p-values FWE-corrected). The effect for pride ratings survived FWE-correction on the whole-brain level (−2, 56, 0, t(38) = 6.15, Z = 5.09, k = 24, p = 0.008), while the one for happiness did not. However, directly contrasting the parametric modulators did not yield significant results (t(38) = 4.83, Z = 4.24, p = 0.189, FWE-corrected inside the SELF ∩ VALUE mask; see Supplementary Tables 23–28 for parametric modulation effects). This is very much in line with previous literature highlighting the involvement of vmPFC activity in the experience of positive affect[54], here demonstrated for pride and happiness.

Psychophysiological interaction (PPI) analysis showed that when task outcomes were received in the HC condition as compared with the MC condition, functional connectivity increased between the left VS and vmPFC (6, 48, −14, t(37) = 3.80; Z = 3.47, k = 48, p = 0.017; corrected within SELF ∩ VALUE mask). Though more spatially extended in the CMS, this effect partly converged with the additive effects of success and control as reported above. Beyond that, the left VS BOLD signal was more strongly associated with responses in the left angular gyrus (AG) during outcome presentation in the HC condition as compared with the MC condition (−46, −70, 38, t(37) = 5.64; Z = 4.76, k = 24, p = 0.001 FWE-corrected inside the SELF¬VALUE mask). The equivalent analyses for the right VS showed significant effects in left superior frontal gyrus (−18, 38, 46, t(38) = 4.59; Z = 4.07, k = 4, p = 0.028, FWE-corrected inside the SELF¬VALUE mask), but not in vmPFC (4, 38, −4, t(38) = 3.23; Z = 3.02, p = 0.069, FWE-corrected inside the SELF ∩ VALUE mask; see Supplementary Tables 29–33 for PPI effects). In a series of control analyses, we further verified that dynamics in functional connectivity were neither specifically driven by WIN or noWIN outcomes and PPI effects in fact result from increased positive functional connectivity in the HC condition (see Supplementary Tables 34 and 35; Supplementary Fig. 2).

Based on the assumption that stronger integration of self- and outcome-related processing should relate to more pronounced pride responses, we performed exploratory analyses on the connectivity dynamics induced in the context of internal control beliefs. These showed that increased connectivity with the VS correlated significantly with the behavioral pride response, at least in the left AG (left AG: Pearson's r = 0.37, p = 0.032; dmPFC: r = 0.23, p = 0.080; precuneus: r = 0.23, p = 0.080, FDR-corrected one-sided p-values; Fig. 6, see "Methods"). Equivalent analyses for the happiness response did not show significant effects (left AG: r = 0.08, p = 0.582; dmPFC: r = 0.02, p = 0.582; precuneus: r = −0.03, p = 0.582; FDR-corrected one-sided p-values). However, we note that these correlations did not differ significantly between pride and happiness (left AG: Hotelling's t(35) = 1.84, p = 0.088; dmPFC: t(35) = 1.29, p = 0.102; precuneus: t(35) = 1.60, p = 0.088; FDR-corrected one-sided p-values). While general positive

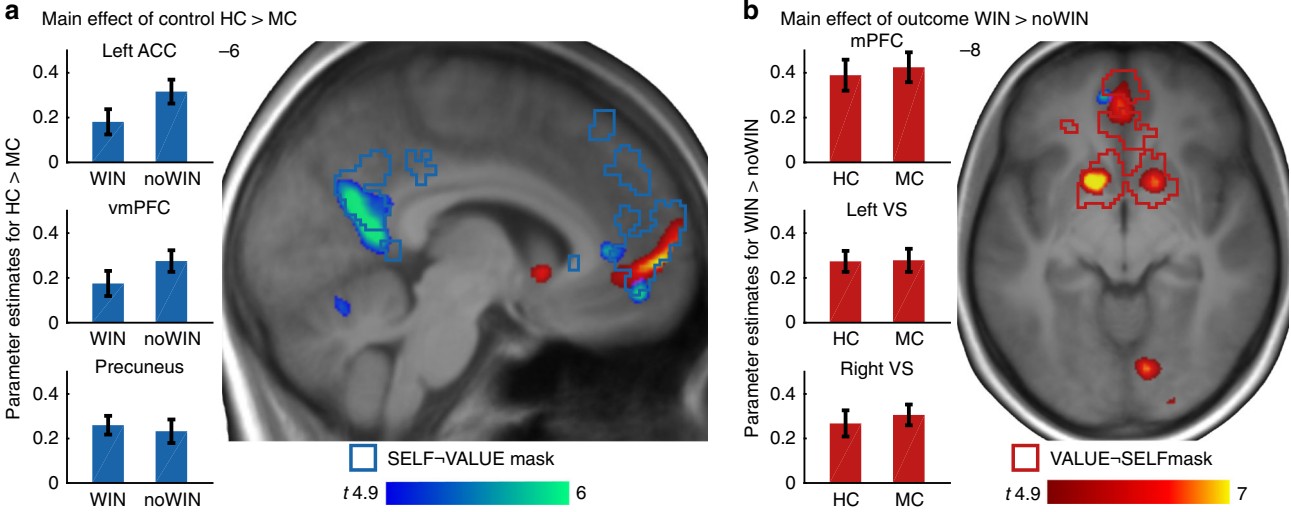

**Fig. 4 BOLD effects of the MRI task. a** HC > MC (blue–green) and **b** WIN > noWIN (red–yellow) displayed at $p < 0.05$, whole-brain FWE-corrected. Blue and red outlines represent masks for SELF¬VALUE and VALUE¬SELF, respectively. Bar plots on the left represent average effects across participants for all voxels inside HC > MC clusters. Bar plots on the right depict average effects across participants for all voxels inside WIN > noWIN clusters. Error bars are $+/-$ 1 standard error of the mean. Source data are provided as a Source Data file.

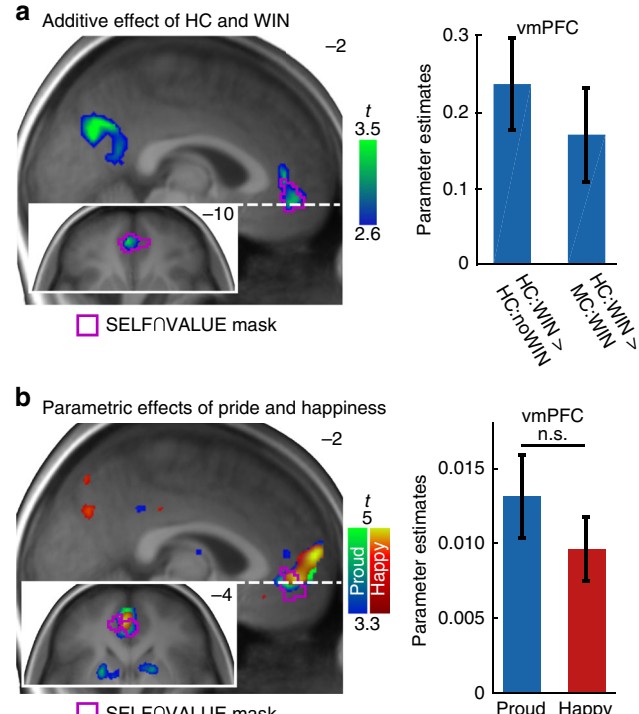

**Fig. 5 Winning under high control and parametric effects of emotion. a** Activations show the additive effect of success and control with the two-way conjunction [HC:WIN > HC:noWIN] ∩ [HC:WIN > MC:WIN] displayed at $p < 0.005$, uncorrected. The conjunction contrast survived small-volume FWE-correction at $p < 0.05$ inside the SELF ∩ VALUE mask in vmPFC. Barplot represents differences of neural activation in vmPFC between HC: WIN and the other two outcomes. **b** Parametric modulation effect of happiness (red–yellow) and pride ratings (blue–green) at outcome presentation, displayed at $p < 0.001$, uncorrected for illustrative purposes. Barplot shows group-level parameter estimates of the parametric modulations at the peak of each parametric effect inside the SELF ∩ VALUE mask. Violet outlines show SELF ∩ VALUE mask. Error bars are $+/-$ 1 standard error of the mean. n.s. not significant. Source data are provided as a Source Data file.

affect has recently been shown to relate to neural activity elicited by surprising and uncontrollable task outcomes[32,33], our data demonstrate that positive affect is also shaped by perceived control over task outcomes. Internal control beliefs increase self-relevance of task outcomes, driving activity in CMS and their connectivity with the VS as well as dynamics of self-related affect upon outcome reception[30,52,53].

**Pride and the subjective value of internal control.** In study 3, we aimed to show that participants' preference for controllable tasks is related to their pride response, even at a monetary cost. After completion of the task as presented before, subjects ($N = 50$) performed a choice task that linked HC and MC options to different monetary amounts. We employed an adaptive staircase algorithm thus varying the offers presented for HC and MC on every trial, allowing to identify each subject's decision point. To increase task relevance, subjects had to execute their choice at random intervals (overall eight trials; outcomes were not presented), and were disbursed with the payoff after study completion (see Supplementary Methods for details).

Again, subjects experienced more internal control in HC than in MC ($t(49) = 11.35$, $p < 0.001$, $d = 1.61$, 90% CI = [1.25; 1.95]) and pride reactivity to outcomes differed between HC and MC (condition*outcome valence: $F(1,49) = 35.98$, $p < 0.001$, $\eta_p^2 = 0.42$; Fig. 7). Specifically, pride was rated higher for WIN than for noWIN outcomes (MC: $t(49)=7.15$, one-sided $p < 0.001$; HC: $t(49) = 12.37$, one-sided $p < 0.001$, corrected). In addition, the difference in pride ratings between WIN and noWIN outcomes was larger in the HC condition than in the MC condition ($t(49) = 6.00$, one-sided $p < 0.001$; for full rmANOVA effects see Supplementary Table 4).

In the choice task, the mean percentage of choices in favor of the HC option was 58.1% (SD = 10.2%), which was significantly higher than chance ($t(49) = 5.62$, one-sided $p < 0.001$). The mean value of the chosen MC options was 1.19 € (SD = 0.09 €) and that of the chosen HC options was 1.03 € (SD = 0.07 €; paired $t$ test: $t(49) = 8.20$, one-sided $p < 0.001$), indicating that participants had a preference for the HC task, despite having a 13.4% smaller payoff than for the MC options.

In order to better characterize the choice behavior, we assessed the model parameters of a logistic choice model fit to each

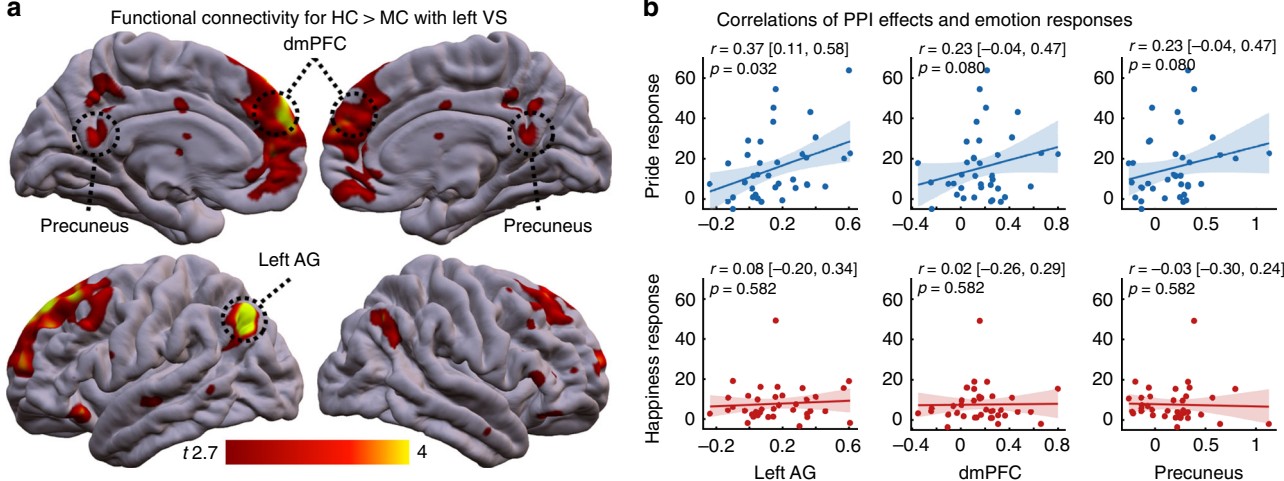

**Fig. 6 Connectivity dynamics of left VS for outcomes obtained under high internal control. a** ROI analysis showed increased coupling between left VS and the left AG, the dmPFC and precuneus (small-volume FWE-corrected at $p < 0.05$). Effects are displayed at $p < 0.005$, whole-brain, for illustrative purposes. See supplements for full connectivity coordinates. **b** Bivariate correlations between pride (top) and happiness (bottom) response (i.e., [HC: WIN–HC:noWIN]– [MC:WIN–MC:noWIN]) and estimates of functional connectivity from left AG, dmPFC, and precuneus. Eigenvariates of clusters within the SELF¬VALUE mask consisting of voxels showing significantly increased functional connectivity with VS at $p < 0.0005$ (uncorrected) were computed for the three regions highlighted in A. $p$-values are one-sided and false-discovery rate corrected. $r$ = Pearson correlation coefficient. Values in brackets are 90% confidence intervals, and shaded areas represent 95% confidence intervals. Source data are provided as a Source Data file.

participant's data (see "Methods"). Across participants, the mean value of the model's intercept $\beta_0$ was significantly negative ($M = -3.49$, SD = 4.18, $t(49) = -6.00$, one-sided $p < 0.001$) indicating that participants preferred the HC option over the MC option, if both options offered equal monetary payoffs (with over 90% of the participants preferring the controllable over the less controllable option: MD = 95.1%, IQR = 20.21). Similarly, $\beta_1$ was significantly negative ($M = -17.21$, SD = 19.67; $t(49) = -6.19$, one-sided $p < 0.001$), showing that across participants the probability for choosing HC increased with higher monetary gain for HC relative to MC (Fig. 7). Further analyses supported the preference of HC over MC offers. First, the monetary value of control, i.e., the median difference in money offered for MC and HC on each trial (see "Methods") was significantly positive, indicating higher offers for MC than for HC (Wilcoxon signed-rank $W = 1199$, one-sided $p < 0.001$, MD of difference = 0.23 €, IQR = 0.20) and participants' willingness to forgo money to execute control. Second, the point of equivalence (POE), i.e., the (log-transformed) value of HC relative to MC at which subjects were equally likely to choose either option, was also significantly negative (Wilcoxon signed-rank $W = 54$, one-sided $p < 0.001$, MD = $-0.29$, IQR = 0.36; see Supplementary Table 6). Importantly, the subjects' probability to choose HC given equal expected values for HC and MC remained significantly above 50% (MD = 65.98%, IQR = 49.92, Wilcoxon signed-rank $W = 864$, one-sided $p = 0.015$) even when considering differences in subjectively perceived outcome probabilities between the two conditions, similar to the POE (Wilcoxon signed-rank $W = 283$, $p < 0.001$, MD = $-0.13$, IQR = 0.37, see "Methods").

Further, participants with a stronger pride response had a higher probability to choose HC in the choice task, given equal values of the two options (Spearman's rho = 0.47, one-sided $p < 0.001$; Fig. 7) and were also more willing to forgo money in order to play the HC task (Spearman's rho = 0.29, one-sided $p = 0.029$; for POE this effect was not significant after correcting for multiple comparisons; Spearman's rho = $-0.18$, one-sided $p = 0.106$; all $p$-values FDR-corrected). In line with the previous literature on perceived control and task preferences[3,55], we also observed that people who experienced stronger differences in

control beliefs (HC-MC) more strongly preferred the HC task. Participants experiencing stronger control were more likely to select the HC task in case of equal offers (Spearman's rho = 0.28, one-sided $p = 0.048$), were willing to forgo money in order to play the HC task (Spearman's rho = 0.26, one-sided $p = 0.048$), and showed shifts in the POE so that MC offers needed to be larger in order to achieve similar probabilities of selecting HC and MC (Spearman's rho = $-0.24$, one-sided $p = 0.048$, all $p$-values FDR-corrected; see Supplementary Table 7). Collectively, these findings support and extend previous studies that showed that mere choice is preferable to not having choices[18,55] by highlighting the notion of instrumental control over the outcomes for building preferences[23], and emphasizing that self-related positive affect might be a relevant factor for mediating motivated behavior[25].

Aggregating the data from all 129 participants, we finally tested whether interindividual differences in the degree to which the task could stimulate control beliefs were associated with self-related affective experiences. Across the three studies, participants who perceived more control in HC compared with MC also showed stronger pride responses (Spearman's rho = 0.28, one-sided $p = 0.001$; Fig. 8), supporting the notion of how individual differences in the experience of internal control relate to self-conscious positive affect[24,30].

## Discussion

Our studies demonstrate that task environments that offer individuals the opportunity to employ internal control beliefs entail unique affective consequences with associated shifts in neural processing. As predicted by theory, we observe greater construal of pride when receiving task outcomes. This aligns with increased activity in brain regions of the cortical midline that are associated with self-related processing[52,53]. Here, the vmPFC seems to play a key role in tracking success, internal control beliefs, and affective dynamics. In addition, activity in these regions is more strongly related to the VS when outcomes depend on one's own behavioral performance. Last, we highlight links between the subjective value of control and emerging positive affect, as people discount

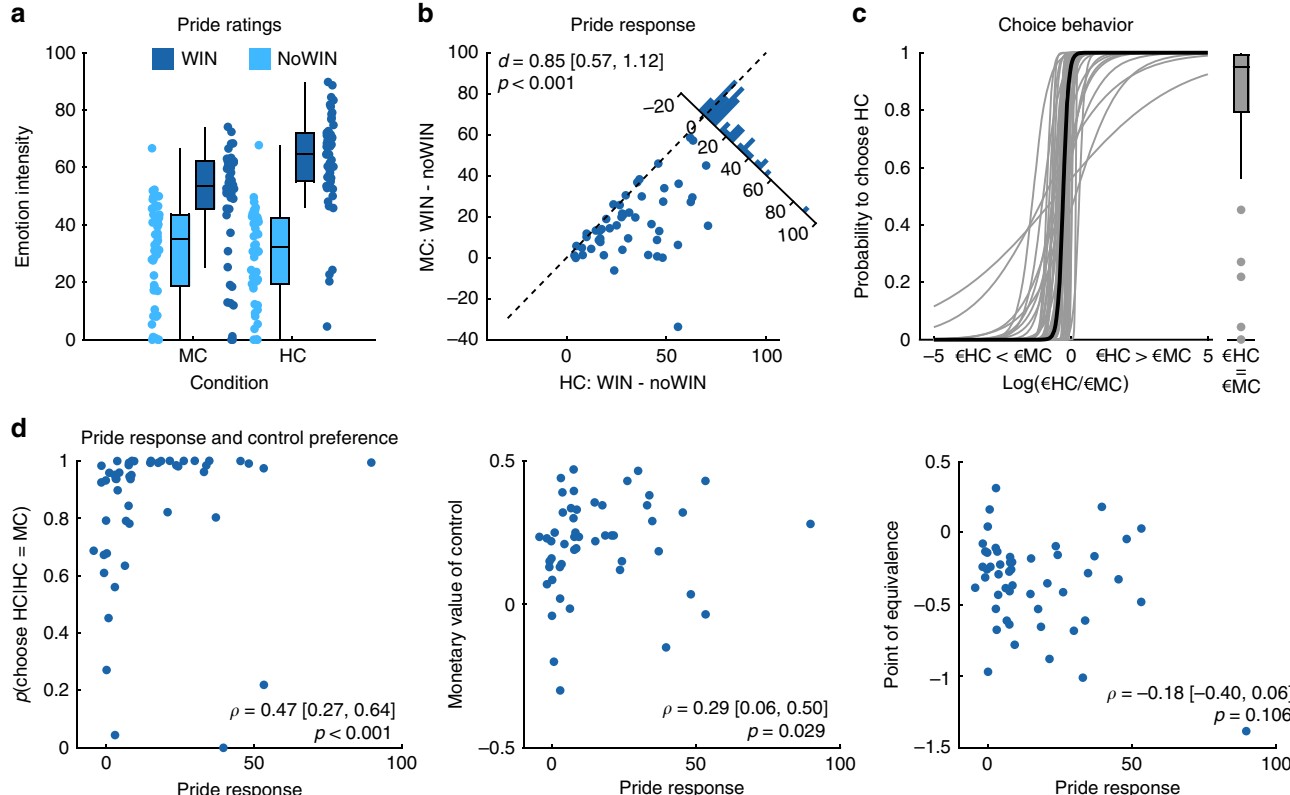

**Fig. 7 The experience of pride in response to high control affects subjective value of choices. a** Pride ratings for study 3, separate for high and low control, as well as outcome valence. **b** Pride reactivity to WIN vs noWIN outcomes, separately for HC (x-axis) and MC (y-axis). Histogram depicts difference between HC and MC. $d$ = Cohen's d. Values in brackets represent lower and upper bound of 90% confidence interval for d. **c** Probability to choose HC in choice task of study 3 as a logistic function of log-transformed relative option values (log(€HC/€MC)). More positive values on the horizontal axis indicate higher monetary value of HC relative to MC. Thin gray lines represent individual participants' choice functions. Thick black line represents hypothetical choice behavior based on sample medians for $\beta_0$ and $\beta_1$. Boxplot represents participant' probabilities to choose HC, given that both options have equal values. **d** Relationship of pride response and choice behavior in study 3. $\rho$ = Spearman's rho. Values in brackets are lower and upper bound of 90% confidence interval for rho. p-values in bottom row are one-sided and FDR-corrected. The center line of boxplots depicts the median, the upper, and lower box borders are the 25th and 75th percentile, and the length of the whiskers is 1.5 times the interquartile range. Source data are provided as a Source Data file.

monetary gains for tasks offering internal control in dependence of their pride response during task execution.

Overall, our findings concord with psychological theories of self-conscious emotions[24] and appraisal theories[27,28], stating that internal control beliefs allow attributing task outcomes to internal causes. Outcomes obtained under internal control beliefs are informative about one's abilities[56] and participants can use these outcomes to infer to what extent they succeed at actually exerting control. The ensuing self-evaluative processes are crucial to self-conscious emotions like pride[24], differentiating them from other emotion constructs like happiness[32]. In this study, this was reflected by a qualitative shift of affective ratings toward pride together with a general increase in positive affect. This supports the hypothesis that control beliefs are not merely related to positive valence but also contribute to a qualitatively distinct construal of affect when outcomes are obtained[24,27]. Here, individuals with stronger pride responses also preferred the task offering more internal control over the less controllable one. This concords with earlier findings on the value of control[18,55,57], critically substantiating the claim that self-conscious affect guides future behavior[24] and illustrating motivational effects of pride[25,29]. The preference to exert control even at monetary costs indicates that behavior is partially driven by the extent to which an individual perceives contexts to be controllable and how

mastering such contexts shapes positive affect, dovetailing nicely with theories linking experiences of competence to intrinsic motivation[1].

The effects of internal control beliefs on outcome processing are mediated via two distinctive, but connected brain networks associated with the processing of self-related information and value. Outcomes from tasks that offered the opportunity to exercise control are inherently self-relevant and were associated with activity in brain regions linked to self-related information processing, such as CMS[41,42]. These findings thus are in line with other studies showing that the ACC is implicated in monitoring self-initiated actions[58] or the vmPFC being involved in outcome monitoring, especially when events are relevant to the individual[41]. In addition, the vmPFC was responsive to the valence of outcomes and showed increased functional coupling with the VS in the context of internal control beliefs. Thus, the vmPFC might play a critical role for integrating self-relevance and outcome valence. The observed covariation of activity in the vmPFC with self-reported affect thereby concurs with previous ideas on how the vmPFC supports the affective valuation of outcomes in more or less controllable conditions[59]. Overall, our results support the notion that self-conscious affect depends on the integration of outcome information with self-representations, mediated via regions of the cortical midline[52].

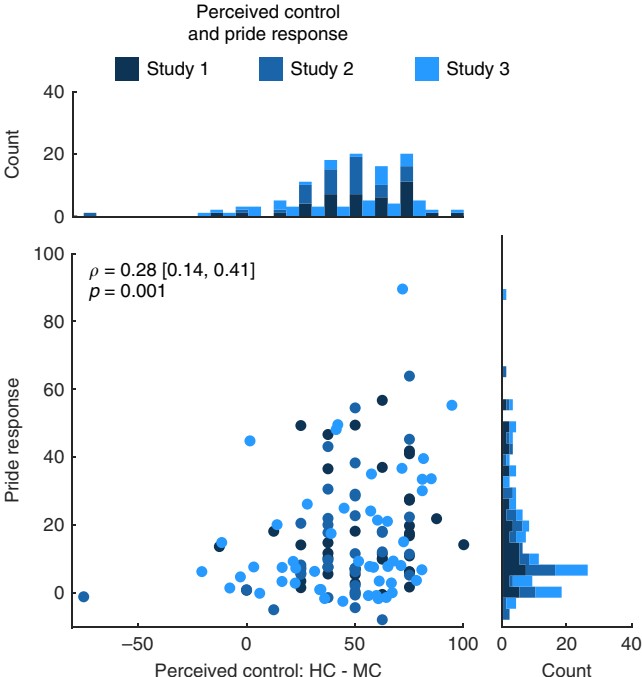

**Fig. 8 Pride and perceived control.** The scatterplot depicts the relationship between internal control beliefs and pride response across all three studies. Dark blue indicates data point belongs to study 1, intermediate blue to study 2, and light blue to study 3. Raw controllability ratings were transformed to a range between 0 and 100, across internal control conditions, before computing the difference between HC and MC in each study. For study 1, raw control and pride ratings were averaged across LC and MC. The pride response is the interaction term for pride ([HC:WIN–HC:noWIN]–[MC: WIN–MC:noWIN]). Histograms display the difference in perceived control (top), and the pride response (right). $\rho$ = Spearman's rho. Values in brackets represent lower and upper bounds of the 90% confidence interval for $\rho$. Source data are provided as a Source Data file.

Regarding the relationship of affect and neural processing, Rutledge et al.[32] recently linked moment-to-moment variability of happiness to reward prediction signals in VS. In a follow-up study, they highlighted the importance of investigating instrumental control for a deeper understanding of well-being[33]. Our study takes a step in this direction by manipulating internal control beliefs, underlining their importance for affective experiences and downstream motivational effects that manifest in behavioral preferences. While recent work suggests how control beliefs and prediction error processing might interact[17,23,50], explicit neuroscientific investigations of their interplay are scarce. Initial evidence suggests that the processing of probabilities in vmPFC, but not VS, can be distorted by illusory control beliefs[59], which reflect the integration of control beliefs and outcome valence in the vmPFC observed here. However, future studies should directly test how prediction error signaling changes in controllable environments, how this relates to affective dynamics, and how these processes are mediated neurally[33,60].

Pride experiences in response to outcomes that can be attributed to own behavior, abilities, or efforts foster self-esteem. Thereby they effectively shape meta-cognitive beliefs about oneself and estimates of one's social status[31,61]. Various studies show the that vmPFC is responsive to social evaluative feedback, especially in individuals with low self-esteem[62], and serves self-protection in terms of more positive self-evaluations when individuals are faced with negative social feedback[48]. In this line, updating of state self-esteem in response to evaluation through

others was found to be positively correlated with activity in vmPFC[46] and thoughts about pride-associated events resulted in significant activations in this region[63]. Furthermore, several studies have supported the role of the vmPFC in updating self-related beliefs[45,47,64]. For instance, the mPFC was related to participants' belief updating regarding their desirability as a person in response to social feedback[47]. From this broader perspective, the vmPFC activation and affective dynamics we observed in response to outcomes from controllable tasks might reflect correlates of short-term updating of beliefs about one's abilities. If subjects experience pride this might contribute to updating their concepts of their own capabilities in a specific environment. In the long run, such positive self-evaluations support self-esteem and help individuals with developing an understanding of their worthiness[65].

The findings described here have implications for learned helplessness and psychiatric conditions, such as major depressive disorder[12,38]. Lower internal control beliefs are associated with reduced attribution of positive outcomes to the self, relating to decreased pride experiences. Further, these reductions in self-related affective responses align with diminished preferences for tasks offering opportunities for exerting control. These dependencies might contribute to diminished self-esteem in the long run[31], and lowered motivation to show goal-directed actions[1,12]. Such processes could therefore causally relate to depressive episodes, which are centrally defined by negative views of the self and lack of motivation[66]. In contrast to states of reduced internal control, other psychiatric phenomena such as pathological gambling are characterized by an increase in internal control beliefs, even in contexts offering no evidence that own actions are linked to outcomes by ways other than chance, a so-called illusion of control[67–69]. Individuals who perceive control over objectively uncontrollable events might experience positive self-related affect once a desired outcome—such as cracking a one-armed bandit—is obtained, and might thus be at greater risk of developing gambling addictions.

In conclusion, the three studies provide converging evidence that internal control beliefs impact the valuation of outcomes, resulting in entangled positive self-conscious affect and behavioral preference for environments with greater perceived control. Based on classic theories of control beliefs[4,6,7], these studies add to our understanding of how outcomes that have been achieved through one's own actions recruit neural systems involved in computations of reward value and self-reference[53]. Increased interactions of the ventral striatum and cortical midline structures, as well as responses of the vmPFC that track both shifts in outcome processing under internal control beliefs and self-conscious positive affect thereby provide links to how self-attribution of desirable events drives motivation and future behavior[23]. On a more general note, our studies underline the need for the neurosciences to consider how behavior and subjective experiences are shaped by control beliefs individuals hold, that regard how their actions relate to events in the world they live in and interact with. Taking further steps in this direction promises to deepen our understanding of affect, motivation, and underlying neural dynamics with important implications for scientific models of behavioral control, general well-being, and psychiatry[1,2,23,50].

## Methods

**Participants.** A total of 154 participants took part in the three studies. For a total of 129 participants valid data were available and included in the analyses (see Supplementary Table 1; mean age: 23.41 years; SD: 3.41; range, 18–37; 77 females). All experimental designs complied with all relevant ethical regulations for work with human participants and were approved by the ethics committee of the University of Lübeck, Germany (AZ 16-133). Participants had to be right-handed, speak fluent German, have normal or corrected-to-normal vision, no deficits in

color vision, and no pre-existing psychiatric or neurological conditions. All participants gave written informed consent, were debriefed after completion of the experimental session, and received monetary compensation or partial course credit for participation.

**Task design and procedure**. In the main experimental task, we manipulated internal control beliefs over task outcomes across three levels (low, medium, and high control; in the following LC, MC, and HC) while keeping the objective probability of achieving positive outcomes at 50% for each level of control. The task displayed a grid pattern of two halves of turquoise and yellow squares surrounding a single gray square in the center (Fig. 1). On each trial, orientation of halves (left–right or upper–lower arrangement) and brightness of squares within each half were pseudorandomly defined. On each trial, three, four, or five squares on each half (targets) were assigned a brightness that was considerably higher than the remaining ones (distractors). Crucially, while there were random differences in brightness of all distractor squares, target squares were designed so that definite identification of the brightest square within one-half was impossible for participants. Various checks verified the validity of the experimental paradigm, including the number of objectively correct choices, subjectively perceived percentages of WIN outcomes, and reaction times (see Supplementary Methods and Supplementary Table 5).

Perceived controllability of task outcomes was manipulated in a within-participants design. In the low control (LC) condition, participants were instructed to click on the central gray square with a computer mouse. In the medium control (MC) condition, participants were instructed to choose one color (i.e., yellow or turquoise) by clicking on any square of that color. In the high control (HC) condition, participants were instructed to identify the brightest square of either the yellow or turquoise half of the grid. For each condition, 50% of the trials were pseudorandomly followed by a reward (WIN) while in the other 50% no reward was given (noWIN). In each study, an equal amount of trials was presented for every combination of outcome valence (WIN, noWIN) and control (LC, MC, HC in study 1; MC and HC in studies 2 and 3). Ratings of pride and/or happiness were repeatedly assessed with equal distribution across combinations of outcome valence and internal control condition (see Supplementary Methods). All experimental stimuli were presented using the Psychophysics Toolbox[70] for Matlab (Mathworks Inc., Natick, MA). Post-experimental interviews were administered using either paper-pencil or computerized questionnaires presented in Sosci Survey (Sosci Survey GmbH, München, Germany). In all three studies, participants completed a practice session prior to the main experiment (see Supplementary Methods) to learn the timing of the task and the outcome probabilities with 50% WIN and 50% noWIN feedbacks in all three conditions.

In the main task of study 1, participants performed a total of 90 trials with 15 WIN and 15 noWIN outcomes at each level of control (see Supplementary Methods for task timing). Ratings of pride and happiness were displayed side-by-side after each trial and visualized as thermometers initiated at 0 and ranging to 100. Participants could rate their subjective feelings of pride and happiness in any order by clicking on the corresponding levels of the thermometers.

Few changes were made in the procedures of the task in study 2. First, the LC condition was not presented in the fMRI in order to reduce scanning time and to perform a more conservative comparison between the MC and HC conditions as both require an element of choice. A total of 80 trials (20 for each combination of condition and outcome valence) were presented, followed by a rating of either pride or happiness (Fig. 1b) on each or every second trial. This yielded a total of 24 ratings each for pride and happiness with 12 for level of control and 6 for every combination of WIN/noWIN and condition. For approximately half of the participants, the ratings of pride and happiness were switched. In case a mistake was committed (i.e., reaction too slow or clicks on the background), both the feedback and the rating on that trial were replaced by a request to follow the task rules.

After a short explanation of the task outside the scanner, the practice session was conducted inside the MRI scanner and followed the same structure as in the study 1, as did the post-experimental interview and debriefing. Participants completed all tasks with an MRI-compatible computer mouse (NAtA TECHNOLOGIES, Coquitlam, Canada) using their right hand which was placed on an armrest with mousepad. ITIs were pseudorandomly jittered between 3 and 5 s during which a fixation cross was presented in the center of the screen. Following a 1 s cue, the task was presented for 4.75 s and after a 4 s delay, the outcome was presented for 3 s. Ratings of the affect in response to the preceding outcomes had to be provided within 4 s on a visual analogue scale, starting at an intermediate position ("somewhat") and ranging from "not at all" to "very" (Fig. 1) corresponding to values from 0 to 100.

Study 3 consisted of two behavioral paradigms (Fig. 1b). After an initial demonstration of the task and a practice session, 64 trials of the main task were presented, and pride ratings were collected on 40 pseudorandomly selected trials equally distributed across the two levels of control and outcome valence (see Supplementary Methods for task timing). Right after the main task, participants completed a choice task in which on each of 96 trials they were given a choice between performing either the MC or the HC condition. Importantly, each choice option was associated with varying amounts of money. For instance, a subject could have the choice between participating in the MC condition for a

potential gain of 1 € or, alternatively, participate in the HC condition for a potential gain of 0.9 €. On eight pseudorandomly selected trials, participants actually performed the chosen task, and were informed that after the experiment they would receive the sum of all successfully completed trials. No outcomes were presented during this part of the experiment to avoid influencing the participants' expectation of succeeding.

The algorithm we used to determine the potential monetary gains of the choice options on each trial followed an adaptive staircase algorithm used in psychophysics. Specifically, on each trial, one of the two choice options was selected as the reference and assigned a value of 1 €, while the other option was assigned another value that was either higher or lower than the reference value. If this comparison value was higher than the reference value on a given trial and the participant picked the higher value, the comparison value was decreased by a predefined fraction in the following trials until the participant changed her decision and chose the option with the reference value. A more detailed description of the staircase algorithm we employed is given in the Supplementary Methods.

**fMRI Data Acquisition**. Participants were scanned using a 3T Siemens MAGENTOM Skyra scanner (Siemens, München, Germany) at the Center of Brain, Behavior, and Metabolism (CBBM) at the University of Lübeck, Germany. After four dummy-scans allowing for equilibration of T1 saturation effects, 768 volumes with 39 near-axial slices in ascending order were acquired for each participant using echo-planar-imaging (voxel size = $3 \times 3 \times 3$ mm, $68 \times 68$ matrix, 20% interslice-gap, TR = 2000 ms, TE = 25 ms, 90° FA, iPAT = 2). In addition, a high-resolution anatomical T1 image was acquired that was used for normalization (voxel size = $1 \times 1 \times 1$ mm, $256 \times 256$ matrix, TR = 1900 ms, TE = 2.44 ms, 9° FA).

**Statistical analyses**. The statistical analyses of subject ratings were performed using JASP version 0.9[71]. MRI data were analyzed using SPM12[72], and the logistic choice models in study 3 were fitted using Matlab R2016a. The comparison of correlations between PPI effects and affective responses in study 2, FDR correction of p-values, as well as the computation of partial rank correlations in study 3 were performed in R[73] (see Supplementary Methods).

For each participant, we calculated the mean ratings of pride or happiness (study 3: only pride), separately for each combination of condition (LC (only study 1), MC, and HC) and outcome valence (WIN, noWIN), for all valid trials (i.e., excluding those in which the participant either clicked on the background or did not respond fast enough). The resulting variables were then taken as dependent variables in repeated measures analyses of variance (rmANOVA), using the factors condition, outcome valence, and affect (factor affect only in studies 1 anfd 2). Paired comparisons were then used to disentangle significant effects of interest.

In order to perform a more direct test regarding the relationship between internal control beliefs and pride, we computed the pride response from the data of each study ([HC:WIN–HC:noWIN]–[MC:WIN–MC:noWIN]). Hence, a positive pride response indicated that participants reported greater outcome-dependent differences in pride under HC than under MC. In addition, after transforming the control ratings from each study to values between 0 and 100 across HC and MC, we computed the difference between control ratings for the different levels of control. For this analysis, control ratings and pride ratings from study 1 were averaged across LC and MC. Nonparametric correlation analysis (Spearman's rho) was used to test the relationship between the pride response and differences in control beliefs between HC and MC.

We predicted the probability for each subject to choose the HC option on a given trial by fitting a logistic model to the choice data. We calculated the ratio of the potential monetary gain for the HC option (e.g., 1.37 €) to the one for the MC option (e.g., 1.01 €) on each of the 96 trials and transformed this ratio to log-space (e.g., log(1.37/1.01) = 0.3049). Thus, positive values of the resulting predictor variable indicate a higher potential gain for HC than for MC on a given choice trial, negative values indicate a relatively higher potential gain for MC, and a value of zero indicate equality of gains for the two options.

Considering the logistic model,

$$p(\text{choose HC}|X) = \frac{1}{1 + \exp(\beta_0 + \beta_1 * X)} \quad (1)$$

participants with parameters $\beta_0$ and $\beta_1$ both equaling zero would have a 50% probability of choosing HC on a given trial, regardless of the value of $X$ ($X$ being the log-transformed ratio of the monetary values offered for HC and MC). Assuming equal offers for HC and MC, negative values of $\beta_0$ indicate that a participant has a preference for HC over MC, while positive values of $\beta_0$ indicate a preference for MC over HC. With more negative values of $\beta_1$, the probability of a participant choosing HC therefore increases more steeply, the higher the potential gain in the HC option, relative to MC.

In order to assess each subject's preference for control, we computed three indices for a preference of HC over MC. First, by setting $\beta_1$ to 0, $\beta_0$ directly defines the probability of choosing HC when monetary offers for HC and MC are equal (p(choose HC|HC=MC); see Fig. 7). Second, we determined the monetary value of control for each subject by computing the median of the trial-by-trial difference in money offered for choosing MC minus the offer for HC on that trial. Third, we computed the point of equivalence (POE), i.e., the (log-transformed) monetary value offered for HC relative to that for MC, at which a given subject was equally

likely to choose either option, i.e., had a probability of 0.5 to choose HC. Thus, we subtracted 0.5 from each subject's logistic function and then used the *fzero* function in Matlab to determine the POE. In this case, a more negative POE indicates a stronger preference for control. We controlled the rank correlation between pride response and each of the three indices for a number of variables regarding the subject's experience of the task, that were rated between main task and choice task (Supplementary Tables 8 and 9).

Participants reported having received WIN outcomes more often in HC than in MC (see Supplementary Table 10), contradicting objective outcome rates. To control for differences in the subjective outcome rates, we again fit the logistic functions to each subjects' choice data, now incorporating subjective outcome probabilities in the prediction. First, each monetary amount offered for either HC or MC on a given trial was multiplied with the subjectively rated percentage of WINs obtained in the respective condition (e.g., 1.23 € × 52% WINs in HC = 0.64 €). The ratio of these expected values on each trial was then log-transformed and used to predict the subjects' choices on each trial.

**Analysis of fMRI data**. fMRI data were analyzed using SPM12[72] in MATLAB R2015a. All functional volumes were slice-time corrected, spatially realigned, and normalized using the forward deformation fields as obtained from the unified segmentation of the anatomical T1 image[72]. Normalized images were resliced with a voxel size of $2 \times 2 \times 2$ mm and smoothed with an 8-mm full-width-at-half-maximum isotropic Gaussian kernel. To remove drift, functional images were high-pass-filtered at 1/128 Hz.

Statistical analyses were performed in a two-level, mixed-effects approach. The first-level general linear model (GLM) for each participant included a total of eight regressors defining the onsets and duration of the two task phases (MC and HC), the four feedback phases (MC:WIN, MC:noWIN, HC:WIN, and HC:noWIN), and two regressors modeling the phases for pride and happiness ratings. In addition, the six realignment parameters from the preprocessing as well as their first derivatives were included as regressors of no interest to account for noise due to head movement. On the second level, we implemented a within-subjects ANOVA, including the first-level contrast images for the MC:WIN, MC:noWIN, HC:WIN, and HC:noWIN outcomes. We computed main effects of condition (HC > MC) and outcome valence (WIN > noWIN), the conjunction (HC:WIN > HC:noWIN) ∩ (HC:WIN > MC:WIN), as well as the interaction ((HC:WIN > HC:noWIN) > (MC:WIN > MC:noWIN)).

In a further GLM analysis, on the first level we additionally modeled the cue phase of each trial, when subjects were informed about which of the two conditions (HC or MC) was presented next. All other regressors were the same as in the previous GLM. The first-level contrast images for the contrast CUE:HC > CUE:MC were then taken to a second-level one-sample *t* test.

A third GLM analysis included two predictors modeling the onsets and durations of the MC and the HC task on each trial, three different predictors for the task outcomes, and two predictors modeling the onsets and durations of the pride and happiness ratings. The first and second of the three outcome predictors modeled those task outcomes after which pride and happiness, were rated, respectively. For each of these predictors the respective ratings on each trial were included as parametric modulators to model associated variability in neural responses. The third outcome regressor modeled those trial outcomes not followed by any affect rating, and thus was not parametrically modulated. As before, the six realignment parameters and their first derivatives were included to account for noise due to head movement. The first-level contrast images for the parametric modulations of pride and happiness were then taken to a group-level within-subjects ANOVA in SPM12[72].

We additionally performed psychophysiological interaction (PPI) analyses on the first level and aggregated the resulting contrast images for the PPI effect on the second level using one-sample *t* tests to investigate functional connectivity between brain regions identified in the within-participants ANOVA. Precisely, we investigated whether functional connectivity of regions activated more strongly for WIN than for noWIN would differ between HC as compared with MC outcomes. For each participant, we defined 4-mm radius spherical ROIs, centered at the nearest local maximum for the contrast WIN > noWIN and located within 6 mm from the group maximum, separately for the left and right VS. By computing the first eigenvariate for all voxels within these ROIs that showed a positive effect for WIN > noWIN, we extracted the time course of activations and constructed PPI terms using the contrast HC > MC[72]. One participant was excluded from the PPI analysis for right VS, because no voxels survived the predefined threshold for eigenvariate extraction. The PPI term, along with the activation time course from the (left or right) VS and a regressor modeling the effect of HC > MC was included in a new GLM for each participant that additionally modeled the two task phases (grids for HC or MC) and rating phases (pride and happiness) as well as the six realignment parameters from preprocessing and their first derivatives.

To associate the PPI effects with the affective experience, we obtained the PPI estimate for the difference in the functional connectivity during MC and HC by extracting their eigenvariate across participants from the three largest significant clusters inside the SELF¬VALUE mask at a more lenient threshold of $p < 0.0005$, uncorrected at peak level. These clusters were located in left AG ($k = 50$ voxels), dmPFC ($k = 89$), and precuneus ($k = 33$; Fig. 6). Next, we correlated pride responses with the PPI estimate of functional connectivity dynamics in each of the three clusters, applying FDR correction for multiple comparisons on the resulting *p*-values. An equivalent analysis was performed for the happiness response. We then compared these correlations between pride and happiness, again applying FDR correction for multiple comparisons (see Supplementary Methods for details on these analyses).

**Reporting summary**. A reporting summary for this article is available as a Supplementary Information file.

## Data availability

The data that support the findings of this study are available from the corresponding author upon reasonable request. The source data underlying Figs. 1–8, Supplementary Fig. 1, as well as Supplementary Tables 2–7 and 9–13 are provided as a Source Data file.

## Code availability

Code used to generate the analyses are available from the corresponding author upon reasonable request.

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

## Acknowledgements

We thank Janine Baumann, Finn Lübber, Timo Schlesinger, and Johanna Schulz for their help with data collection. The research leading to this publication was supported by the German Research Foundation (MU4373/1-1) and the Junior Research Program at University of Lübeck (F.M.P.). We acknowledge financial support by Land Schleswig-Holstein within the funding program Open Access Publikationsfonds.

## Author contributions

D.S.S., F.M.P., L.M.P., and S.K. developed and designed the experiments. D.S.S. programmed the experiments. D.S.S., F.M.P., and S.K. conducted data collection. D.S.S. performed statistical analyses. D.S.S., F.M.P., L.M.P., and S.K. interpreted the results. D.S.S., F.M.P., L.M.P., and S.K. wrote the paper.

## Competing interests

The authors declare no competing interests.
