## [Peer Review File · Nature Communications]

Reviewers' Comments:

Reviewer #1:

Remarks to the Author:

In this paper, the authors conduct 3 experiments to show evidence of increased internal control beliefs lead to increased affective valuation and associated changes in neural processing. They use a task where the level of internal control is modulated while measuring subjects' affect via subjective ratings of happiness and pride. One of the interesting findings is the differentiation between these two conditions, termed general positive affect and more self-conscious positive affect, with the latter being more specific to contexts in which control is experienced. The authors also observed modulation of neural circuits involved in self-processing and valuation, with the vmPFC in particular tracking success and personal control. Finally, they showed that people were willing to discount money to attain control. Taken together, it is a nice collection of cleverly designed studies packaged in a well-written and interesting paper. While I liked the paper overall, there are some questions and suggestions below that merit discussion.

1- The tasks involve the manipulation of perception of control and argues that it is not enough to merely have choice to have self-evaluative affect. I am in agreement with the statement that having choice is not enough. However, this is not inconsistent with prior work cited by the author. Such work tends to compare free-choice vs forced (or random) choice options, and has observed that having the perception of choice is linked to positive affect and neural signals associated with reward and motivation. The current paper goes a step further with a more elaborate comparison while removing the forced/random choice (in Exps 2 and 3, LC in Exp 1). In this way, it is a parametric modulation of context where the more control one perceives the more these affective signals are represented. If the authors' paradigm involved only the MC and LC condition, for instance, it is likely that the differences between MC and LC would be magnified. Thus, while I am in agreement with the authors' premise, I do not fully agree that it is inconsistent with prior work, merely that it is a nice extension showing further contextual differences (with nice added features with regards to relation to self-conscious positive affect).

2- In relation to the last point, it would be helpful to clarify the nature of the MC task. Were subjects requested only to pick between two colors for a 50/50 guess? That is, was it a slight step above the LC condition which involved pressing one button for a 50/50 guess?

3- The task includes the instruction screen highlighting which condition it is. It would be useful for the authors to analyze brain responses to that screen that signify what type of trial it will be.

4- Task timing appears to be missing on all experiments, but particularly the fMRI task.

5- With respect to the fMRI analyses, there are a few questions.

a. First, the 3-way conjunction analysis is unclear and not justified. The contrast of HC Win > MC no Win is not appropriate and it would seem that a better analysis would be an interaction between (HC and MC) (win, no win).

b. Second, the PPI results are interesting. The main result involves increased connectivity in VS and VMPFC as a function of receipt of task outcomes in the HC. Was this all outcomes? Or a contrast of win vs no win? Basically, is it driven by win or no win?

c. Third, also on the PPI, it would be useful to perform the same PPI with MC trials to see if the VS-MPFC connectivity is also there.

6- For the fMRI design, were ratings evenly distributed across wins and no wins (and conditions?) That is, would HC pride trials randomly follow more wins than no win trials for example? Or was the order of subjective ratings also sampled in a way that ensured even distribution across outcomes and conditions?

7- Are happiness and pride correlated in the experiments?

8- For Exp.3 (and to a degree the vmPFC result in 2), it would be useful to discuss Wang et al., 2019, *Cerebral Cortex*, which also showed that individuals were willing to give up value in order to attain control.

Reviewer #2:

Remarks to the Author:

This manuscript reports 3 experiments that evaluate the relationship between affective reactions to outcomes associated with varying levels of control. A novel choice task was used to manipulate control (1. no choice/control (LC), 2. binary choice/gambles (MC), and 3. perceptual discrimination task that is ostensibly ability-based (HC)) and pride and happiness ratings are evaluated after outcomes (Win vs. No-Win). Manipulation checks show that subjective ratings of control increase with the levels of the task manipulation. In Study 1, happiness and pride increased following wins than no wins, but more so as controllability increased, and more so for pride than happiness. In Study 2, value-based regions of the brain (vmPFC, striatum) responded more strongly to wins than no-wins, self-processing regions (vmPFC, ACC, precuneus) responded more strongly to HC>MC conditions, and vmPFC related to both value and self-processing responded more strongly to wins in high control conditions and to pride more so than happiness. Moreover, the vmPFC region showed higher covariation with striatum in the HC condition. Finally, in Study 3, individuals were willing to forgo monetary reward to select HC choices over MC choices, and their propensity to choose HC over MC (when their value was equivalent) correlated to pride responses. The task is clever and well-designed to manipulate perceived control, and the topic is timely and interesting. It is also commendable that the authors pursued their questions across a few experiments with independent samples. However, I have a number of concerns that limit the interpretation and impact of the results, which I outline below:

Given that pride and happiness and importantly their difference is so critical to the current interpretation, it would be important to evaluate the extent to which pride and happiness judgments are correlated. Based on the nature of the task – making perceptual discrimination judgments about luminance and receiving manipulated feedback – it could be that people are experiencing pride and happiness, or some mildly positive response that straddles the two. If people are asked to assign a number to pride and to happiness, they'll do so, but it is unclear if the experience itself represents pride and happiness and whether these are separable.

For instance, the difference in pride between Win and No Win is more influenced by the control manipulation (HC > MC) than happiness, which could lead to the interpretation that "the resulting pride response was significantly larger than the happiness response". But it is the difference of differences that differs by the control manipulation, and not the overall degree of pride or happiness being experienced. The levels of happiness seem larger overall than levels of pride, and the difference between Win and NoWin in HC condition in happiness seems equivalent (Study 2) if not larger (Study 1) than that for pride.

For the HC condition, if feedback was pre-determined, then the actual outcomes are still chance based. If pride depends upon feelings of mastering a task, then are people really feeling pride if the outcomes do not depend on mastery or ability? Did participants suspect that feedback was not tied to performance, as their performance on HC trials could not improve over time? This sense of improvement seems important for mastery and hence pride.

In the fMRI analyses, self-value mask removes value regions from the self ROIs. However, part of the argument being made is that the high control condition is subjectively valuable compared to the other conditions, and indeed the findings from study 3 show this. What is the rationale behind conducting the HC>MC analysis only in self regions rather than valuation (or a combination of both) given the subjective value involved in greater degree of control?

The logic behind the conjunction of value of winning under high control is confusing. Why not simply run a conjunction of HC W > HC NoW and HC W > MC W, or an interaction between HC > MC and W > NoW? The prediction is that Winning under HC will generate larger responses in valuation and/or self-processing regions, which is a spreading interaction. The HC W > MC NoW contrast in the conjunction, to me, does not seem to add meaningful information in the conjunction, other than possibly help increase the effect in vMPFC.

In order to test whether vMPFC activity tracks pride more strongly than happiness ratings, why not conduct a contrast of pride > happiness, which is consistent with the rest of the analyses presented in the manuscript? The analysis in its current form compares beta weights for pride vs. happiness in the Self and Value ROIs. Alternatively, the effects of Winning and Control could also be tested using this approach, by extract beta weights from the ROIs and test the effects of Winning (W/NoW) and Control (HC/MC) rather than running contrasts. The differences in analytic approach raises questions about the robustness of the effect.

For PPI analyses, it wasn't clear whether they tested for connectivity differences (e.g., between striatum and vMPFC) between conditions HC > MC, or whether additionally there were significant PPI effects within the HC condition itself. A significant difference in connectivity doesn't address whether there is a significant connectivity effect in the HC condition itself, just that it is different across conditions.

In study 3, it makes sense that people should choose HC more often if the value is equal with MC, and that this should be related to pride. But the results also show that participants forgo monetary reward to choose HC (average value per trial for HC and MC, probably also in aggregate money forgone during the duration of the task). Shouldn't the degree to which an individual values control (and therefore gives up money to experience HC) relate to pride? This could be tested with the mean value of trials, money forgone during the task, or a point of subjective equality (the value at which people are equally likely to choose HC and MC, which will veer towards HC). It would be good to know if pride correlates with the bias towards HC over MC when their values are equal, and/or whether pride correlates with the propensity to forgo rewards to choose HC over MC.

Minor points:

Related to my comment related to the relationship between pride and happiness, the abstract discusses the findings in relation to affective dynamics or positive affect, but the remainder of the results draw strong distinctions between pride and happiness. Why is there such a discrepancy in the description of the results across the manuscript?

There is a broader literature on self and value (e.g., Somerville et al., JoCN, 2010; Hughes & Beer, JoCN, 2013; Korn et al., JNeuro 2012) that could further support the connection between vMPFC responses to self and reward.

There is also an interesting experiment on reward responses associated with Illusory control that could be included given its relevance to the current study (Kool, Getz, Botvinick, JoCN, 2013).

The abstract introduces the term "self-evaluative affective dynamics" but this is very difficult to understand without a description.

**Revision of Manuscript “Internal control beliefs shape positive affect, motivation, and neural dynamics during outcome valuation”
(Response to Reviews – Manuscript #: NCOMMS-19-18929)**

We are grateful to the editor and the reviewers for their constructive comments as well as their overall recognition of the quality of the reported studies. The suggestions raised in the reviews were very helpful. We very much appreciate the opportunity to revise our manuscript and have addressed all comments raised by the reviewers.

Specifically, we have clarified the exact nature of the experimental task and its distinction to related paradigms that have been previously used in the study of behavioral control. We have extensively analyzed and discussed the rationale and implications of the association between happiness and pride and more clearly delineate why we believe that our task and findings allow to learn novel aspects with regards to behavioral control, outcome valuation, and their relation to affective dynamics.

Furthermore, we performed a series of additional behavioral and fMRI analyses that were suggested by the reviewers, removed or corrected analyses that were rightfully criticized, and included information that was missing. We have also included an additional analysis that was motivated from the suggestions of the reviewers and we considered valuable to better align our results to existing literature on the value of control. In addition, we have broadened our discussion based on the references to the relevant articles the reviewers suggested to incorporate.

We marked all changes to the draft in red and provide a point-by-point reply to the constructive feedback of the reviewers below.

Reviewers' comments:

Reviewer #1 (Remarks to the Author):

In this paper, the authors conduct 3 experiments to show evidence of increased internal control beliefs lead to increased affective valuation and associated changes in neural processing. They use a task where the level of internal control is modulated while measuring subjects' affect via subjective ratings of happiness and pride. One of the interesting findings is the differentiation between these two conditions, termed general positive affect and more self-conscious positive affect, with the latter being more specific to contexts in which control is experienced. The authors also observed modulation of neural circuits involved in self-processing and valuation, with the vmPFC in particular tracking success and personal control. Finally, they showed that people were willing to discount money to attain control. Taken together, it is a nice collection of cleverly designed studies packaged in a well-written and interesting paper. While I liked the paper overall, there are some questions and suggestions below that merit discussion.

Thank you very much for your appreciation of our work and the constructive feedback. We think that your comments have helped us to more clearly elaborate on the novel aspects our studies add to the literature of control beliefs and improved the methods and analytical procedures. Please find the point-by-point reply to your specific comments below.

Comment #1: The tasks involve the manipulation of perception of control and argues that it is not enough to merely have choice to have self-evaluative affect. I am in agreement with the statement that having choice is not enough. However, this is not inconsistent with prior work cited by the author. Such work tends to compare free-choice vs forced (or random) choice options, and has observed that having the perception of choice is linked to positive affect and neural signals associated with reward and motivation. The current paper goes a step further with a more elaborate comparison while removing the forced/random choice (in Exps 2 and 3, LC in Exp 1). In this way, it is a parametric modulation of context where the more control one perceives the more these affective signals are represented. If the authors' paradigm involved only the MC and LC condition, for instance, it is likely that the differences between MC and LC would be magnified. Thus, while I am in agreement with the authors' premise, I do not fully agree that it is inconsistent with prior work, merely that it is a nice extension showing further contextual differences (with nice added features with regards to relation to self-conscious positive affect.

We emphatically agree with the reviewer that our data and conclusions are not inconsistent with prior work. In fact, the neuroscience studies we referenced (e.g. Cockburn, Collins, & Frank, 2014; Leotti & Delgado, 2011, 2014) are central to the understanding of psychological concepts such as choice and behavioral control as well as their underlying neural mechanisms and we agree that this study extends these previous findings. This relevance is also reflected in study 1, which shows that perceived control was higher for MC than for LC (see figure 1D), demonstrating the fundamental role that choices play in perceived control. Thus, we note that despite the presence of the HC condition in study 1, there are still discernible differences regarding the level of perceived control in MC and LC. We very much agree with the reviewer that a mere comparison of MC and LC could likely amplify these differences in perceived control and the available conditions of the particular experiment to shape how participants use the rating scale and anchor their judgment. That being said, in the following, we would like to (i) elaborate what we think is the important qualitative distinction in the HC condition of the present paradigm that is important for investigating novel aspects of perceived control and self-conscious affect, (ii) show some empirical evidence against the idea that the mere experimental

context defines the rescaling of control ratings, and (iii) discuss specificities in how we measure the construct of control that should contribute to the observations.

First, the key difference between LC and MC is having a choice or not, which is comparable to free-choice vs forced-choice paradigms (Cockburn et al., 2014; Leotti & Delgado, 2011, 2014). In contrast, both MC and HC include a facet of choice. Importantly, however, choices in the HC condition are informed choices. That is, individuals will exert effort to collect information needed to identify the brightest square (i.e. through comparison of varying levels of brightness) and then make a choice based on this information. In case they perform sufficiently well, they will receive a WIN outcome, and otherwise a noWIN outcome. This allegedly renders HC outcomes informative about the individuals' efforts or abilities in the experimental context, which lays the basis for attributional processes, the evaluation of (aspects of) the self, and hence the experience self-conscious affect (Tracy & Robins, 2004). Based on this reasoning, we think that it is unlikely that a mere comparison of forced-choice (i.e. LC) vs free-choice (i.e. MC) conditions will render the latter *qualitatively* more similar to HC, even though quantitative differences in control ratings between MC and LC might in fact be amplified when no HC task is available for comparison. This is because in contrast to HC, choices in MC are fundamentally uninformed, and hence, outcomes are only attributable to chance.

Second, in part our data also speak against the idea of contextual rescaling of control ratings. To illustrate this point, we think it is worth comparing effect sizes across study 1 and study 2 + 3 that implement different experimental context with regards to how participants might allocate their perceived control to the different conditions. Here, the mere presence of the LC condition in study 1 does not change the effect size for the difference in perceived control between MC and HC across studies (study 1: 1.77, study 2: 1.53; study 3: 1.61). While we very much agree that contextual rescaling of perceived control might be a factor to consider when evaluating the results, these data indicate that the effects might not severely bias our findings.

Third, we want to emphasize that we measured the construct of control with items that reflect the concept of "internal control beliefs" as suggested by theory. In order to estimate internal control beliefs, participants in study 1 and study 2 rated their control experience with regard to how well they thought they were able to influence the results ("How strongly could you influence the outcome in the task?") and in study 3 additionally with respect to how much control they experienced ("How strongly could you control the outcome of the task?"). Both items were highly correlated ($r = .76$) and are thus considered to represent a similar latent variable that we think represents internal control beliefs. This operationalization of the construct of control better reflects the notion of internal control as implied by classical theories and might explain why we find stable differences across experimental contexts, whether the LC condition being present or not.

We want to emphasize that this line of reasoning in our view does not diminish the fundamental relevance of free-choice vs. forced-choice for understanding perceived control, that has been demonstrated before (Cockburn et al., 2014; Leotti & Delgado, 2014; Leotti, Iyengar, & Ochsner, 2010) and does not imply that our results are inconsistent with previous literature. In contrast and as the reviewer suggests, our study should be considered an extension to prior studies and their conceptualization of control beliefs which we think is valuable specifically, but not exclusively, because of its important implications for affect and motivation.

Based on the theoretical considerations above and the operationalization of the construct of control emphasizing internal control beliefs, we have focused on the comparison of HC and MC given the temporal constraints of the fMRI study (study 2) and methodological constraints of the choice paradigm (study 3). Future experimental and theoretical work is nonetheless

needed to delineate more clearly how perceived control and emotional experience is shaped in the contexts of no-choice (LC), free-choice (MC) and informed choice (HC) tasks.

We have now revised the manuscript to make it clearer that our results support previous findings and are an extension to the existing ideas in the literature. We have thus made changes in the introduction, results, and discussion. Further, we have acknowledged and emphasized the relevance of previous findings for the current studies in order to avoid misunderstandings that the present study is inconsistent with previous results. To also better align our findings with the existing literature, we have included additional analyses on the relation of internal control beliefs with choice behavior in study three, which was initially lacking in the manuscript. Below we highlight the most prominent changes that we have made to address your comment.

Introduction:

II. 60-62 "...a region implicated in dopaminergic reward processing (Knutson, Adams, Fong, & Hommer, 2001; O'Doherty, Buchanan, Seymour, & Dolan, 2006; Pagnoni, Zink, Montague, & Berns, 2002). Theory and empirical literature thus very much agree that having the option to choose increases control beliefs and carries intrinsic value."

II.75-80 "Thus, while having choices is the condition necessary for exerting control, it is in principle not sufficient for experiencing the quality of control belief underlying self-attribution of task outcomes as implied by theory (Ajzen, 2002; Bandura, 1977; Rotter, 1966). Only if the context offers the potential for building the belief of internal control over outcomes, it is possible to attribute events to the self, own efforts, and abilities, with broader implications for self-related affect and motivation (Tracy & Robins, 2004; Williams & DeSteno, 2008)."

Results:

II.166-168 "This replicates previous findings showing that choice is essential for the experience of control (Leotti & Delgado, 2011, 2014), but additionally shows that perceived control comparably excels in situations in which only one's own performance and abilities allow to obtain the desired outcomes."

II.388-401 "In order to further embed our data into the existing literature on perceived control and task preferences (Leotti et al., 2010; Wang & Delgado, 2019), we also tested the association between the three indices of control preference and individual differences in experienced control over the different tasks. Results clearly show that people who experienced stronger differences in control beliefs (HC-MC) also more strongly preferred the HC over the MC task. All three variables indicative of control preferences showed significant effects in the expected directions. Precisely, people experiencing stronger control were more likely to select the HC task in case of equal offers (Spearman's $\rho = .28$, one-sided $p = .048$), were willing to forgo money in order to play the HC task (Spearman's $\rho = .26$, one-sided $p = .048$), and showed shifts in the POE so that MC offers needed to be larger in order to achieve similar probabilities of selecting HC and MC (Spearman's $\rho = -.24$, one-sided $p = .048$, all p -values FDR-corrected for multiple comparisons). Collectively, these findings support and extend previous results that showed that mere choice is preferable to not having choices (Leotti & Delgado, 2011; Wang & Delgado, 2019) by highlighting the notion of instrumental control over the outcomes for building preferences (Ly, Wang, Bhanji, & Delgado, 2019), and emphasizing that self-related positive affect might be a relevant factor for mediating motivated behavior (Williams & DeSteno, 2008).

Discussion:

II. 493-495 “Our findings support and extend previous work on the value of control (Leotti & Delgado, 2011; Suzuki, 1997) by demonstrating that participants preferred a controllable task over an uncontrollable one, reflecting differences in subjective value that are likely processed by vmPFC (Bartra, McGuire, & Kable, 2013).”

Comment #2: In relation to the last point, it would be helpful to clarify the nature of the MC task. Were subjects requested only to pick between two colors for a 50/50 guess? That is, was it a slight step above the LC condition which involved pressing one button for a 50/50 guess?

Yes, this is correct. The LC task asked subjects to click on the central gray square in order to receive either a WIN or a noWIN outcome. In the MC task, subjects were instructed that the computer would randomly define one color to be correct before the beginning of each trial and that subjects should then guess which of the two colors they believed was correct. Those two conditions are conceptually nearly identical to the idea of the experimental setups used in previous studies of behavioral control (Leotti & Delgado, 2011, 2014). Thus, while the LC task did not even include any component of choice, the MC task allowed subjects to choose between two options, comparable to choosing heads or tails in a coin-flip. Thus, a comparison of MC vs LC asks about the fundamental prerequisite of control, i.e. having choices (Leotti et al., 2010). However, in distinction to the HC condition, choices in the MC condition were uninformed, i.e. random guesses, and did therefore not reasonably allow to relate outcomes to own abilities or efforts, as argued in more detail above.

Thank you for pointing out that this distinction might not have been clear enough. We have now included a brief clarification regarding the differences between the conditions at the beginning of the results section.

Results:

II. 148-156: “On the lowest level of control, successes (WIN outcomes; 0.20 €) and non-successes (noWIN outcomes; 0.00 €) depended on an automated gamble which was initiated by clicking on the grey square (low control, LC; see figure 1). Comparable to previous studies (Leotti & Delgado, 2011, 2014), participants in the LC condition thus had no option to choose and employ even the lowest level of control. On the medium level of control, WIN and noWIN outcomes allegedly depended on the correct choice between the two colors (medium control, MC) introducing the opportunity to choose in MC. On the highest level of internal control, WIN and noWIN outcomes depended on whether subjects were able to identify the brightest square within the shades of one color (high control, HC) thus allowing the attribution of task outcomes to the self and internal causes.”

Comment #3: The task includes the instruction screen highlighting which condition it is. It would be useful for the authors to analyze brain responses to that screen that signify what type of trial it will be.

We agree with the reviewer that this analysis is interesting, as it would allow to more coherently relate our study to previous findings that showed that anticipating opportunities for choice (as compared to forced-choice) recruits the ventral striatum (e.g. Leotti & Delgado, 2011). If the anticipation of tasks experienced to be more controllable has greater subjective value, anticipation of the HC condition should induce stronger VS activation than anticipating the MC condition. We performed the suggested analysis (CUE:HC > CUE:MC), and did find

significantly stronger activation of both left and right VS (small-volume FWE-corrected inside the VALUE→SELF mask; see supplementary figure S1 and supplementary table S21). This reflects and extends previous findings showing that anticipation of control recruits regions associated with reward anticipation (Knutson et al., 2001; Leotti & Delgado, 2011). We have included references to this analysis in the result section and clarify that these findings very well align with previous work mainly of the group of Delgado et al. (Leotti & Delgado, 2011, 2014).

Results:

II. 262-268: “Prior studies have demonstrated that the anticipation to exert control increases activation in the ventral striatum (Leotti & Delgado, 2011, 2014). We therefore also tested for differential VS activity in our task during the presentation of the task cues, contrasting HC against MC cues. This analysis yielded significant activations in the bilateral VS (left: -12,8,-6, $t(38)=4.73$; $Z=4.16$, $k=26$, $p=.013$; right: 18,4,-6, $t(38)=4.85$, $Z=4.25$, $k=62$, $p=.009$; small-volume FWE-corrected inside the VALUE→SELF mask, see supplementary table S21 and supplementary figure S1), replicating various previous findings of increased striatal activation when environmental information signals the potential for exerting control (Leotti & Delgado, 2011, 2014).”

Methods:

II. 878-881: “In a further GLM analysis, on the first level we additionally modelled the cue phase of each trial, when subjects were informed about which of the two conditions (HC or MC) was presented next. All other regressors were the same as in the previous GLM. The first level contrast images for the contrast CUE:HC > CUE:MC were then taken to a second-level one-sample t-test.

Supplementary figure S1:

Figure S1. Effect of cue phase. **Left** Activation map on the shows stronger activation for presenting cues signaling an upcoming HC task as compared to presenting cues indicating and upcoming MC task. Red outline shows VALUE→SELF mask. Results are displayed at $p<.0005$, uncorrected. **Right** Bars show beta estimates extracted from peaks in left (x,y,z (mm): -12,8,-6) and right (18,4,-6) ventral striatum. Errorbars show +/-1 standard error of the mean.

Supplementary table S21:

Table S21. Paired t-test for CUE:HC>CUE:MC, small-volume corrected inside SELFVALUE, critical T=4.216

set		cluster				peak				MNI coordinates (mm)													
p	c	p (FWE)	p (FDR)	k	p (unc)	p (FWE)	p (FDR)	T	Z	p (unc)	x	y	z										
<.001	3	.023	.448	5	.448	.003	.354	5.26	4.53	<.001	22	12	4										
						.001	.041	62	.014	.009	.354	4.85	4.25	<.001	18	4	-6						
												22	6	-4									
												24	10	-4									
												8	10	-6									
												6	14	-8									
												.036	.721	4.34	3.89	<.001	18	12	0				
												.005	.135	26	.090	.013	.354	4.73	4.16	<.001	-12	8	-6

Comment #4: Task timing appears to be missing on all experiments, but particularly the fMRI task.

It is correct that the task timings were missing for studies 1 and 3 and we have included these in the methods section of the manuscript in the respective subsections. We are very sorry for this mishap. However, regarding the fMRI study, timings were already included in the methods section of the original version of the manuscript, but admittedly, this was perhaps not extremely well indicated in other parts of the manuscript. To make the reference clearer, we have now included a sentence in the caption of figure 1, which indicates that detailed timing information for all studies is included in the methods section. We thought about including timing information in figure 1 itself, but decided not to, because the differing timings for the three studies would have caused the figure to be quite cluttered.

Caption figure 1:

“See methods for detailed timing information of the three studies.”

Methods:

II. 628-633: “On each trial, after a fixation of 3 seconds and a 1 second cue, the task was presented for a total of 4.5 seconds. 3 seconds after task onset, the cursor appeared at a random position over the squares and subjects could respond within 1.5 seconds. If subjects responded correctly, the outcome was presented for 2 seconds, otherwise a warning message followed without delay and informed subjects about their mistake (e.g. if they were too slow or had clicked on the background instead of a square).”

II. 664-669: “On each trial, following a fixation cross (jittered between .75 and 2.75 seconds) and a 1 second cue, the task was presented for a total of 4.5 seconds. The cursor appeared centrally on the screen .5 seconds after task onset and subjects could respond within 4 seconds. If subjects responded correctly, following a 1.5 second delay, the outcome was presented for 3 seconds. Otherwise a warning message followed without delay and informed subjects about their mistake (e.g. if they were too slow or had clicked on the background instead of a square). If a trial contained a rating, this was presented for 4 seconds.”

Comment #5: With respect to the fMRI analyses, there are a few questions.

a. First, the 3-way conjunction analysis is unclear and not justified. The contrast of HC Win > MC no Win is not appropriate and it would seem that a better analysis would be an interaction between (HC and MC) (win, no win).

We thank the reviewer for highlighting this issue, and we apologize for not sufficiently elaborating on our rationale in the original submission. In principle, the analytical strategies for showing regional specific effects in the brain break down into modeling the *additive* or *super-additive effects* of control (i.e. HC) and positive outcomes (i.e. WIN). It thus depends on the specific hypotheses which analyses are adequate and justified. With the conjunction analyses, we have aimed to test the more general – and from our view straight-forward hypotheses - that whether having control (i.e. HC) and receiving positive outcomes (i.e. WIN) would lead to regional specific, *additive effects* in the ventromedial prefrontal cortex. This equates to asking whether WIN outcomes under HC engage the vmPFC more strongly than any of the other outcomes and the specific set of tests included in our conjunction analysis. This is a different and more general model compared to searching for *super-additive effects* testable with an interaction analysis, as suggested by the reviewer. Such an analysis would allow answering the question of whether the integration of HC (vs. MC) and WIN (vs. noWIN) outcomes follows linear but non-additive principles with respect to BOLD activation of a specific brain region. In fact, performing both analyses, and thereby testing the two alternative hypotheses of whether control and successes lead to *additive* or *super-additive effects*, is a valuable and interesting question even though the latter is considerably more specific. Hence, we have decided to perform the interaction analysis as requested by the reviewer, in order to gain a deeper understanding of how the vmPFC integrates both concepts. However, our data shows no support for super-additive effects neither in the vmPFC nor other parts of the brain, even at more lenient thresholds. We also now discuss the effects of the conjunction analyses more closely related to the idea of additive effects for successes and control in the vmPFC.

Aside from suggesting to run an interaction analysis, the reviewer criticized that the contrast HC:WIN>MC:noWIN is not appropriate in the conjunction analysis we performed. We note that in SPM, the statistical significance of a conjunction analysis is limited by the weakest effect, and testing HC:WIN>MC:noWIN in vmPFC should be expected to produce a larger effect than, for instance, HC:WIN>MC:WIN. Thus, we agree with the reviewer that including this contrast in the conjunction might be unnecessary, but would also like to point out that therefore, it did not affect the statistical significance of the conjunction analysis. This is supported by the fact that an updated conjunction analysis which only included the contrasts HC:WIN>HC:noWIN and HC:WIN>MC:WIN (i.e., excluding HC:WIN>MC:noWIN) yielded almost exactly the same results with regard to resulting p-values and T-statistics. We have now adopted the conjunction without HC:WIN>MC:noWIN, however with similar results. Please find the updated figure below and our response to comment #4 of reviewer #2.

Results:

II. 246-261: “Next, we were interested in the additive effects of internal control and winning, by describing specific neural correlates of receiving positive outcomes under high control beliefs. To do so, we composed the two-way conjunction contrasting success achieved under high control to both success achieved under medium control and no success obtained under high control ($[HC:WIN>HC:noWIN] \cap [HC:WIN>MC:WIN]$; figure 5). This analysis revealed that the left vmPFC, specifically, is more strongly engaged when subjects’ successes are attributable to their own actions (-4,48,-10, $t(114)=3.45$; $Z=3.36$, $k=12$, $p=.022$, FWE-corrected within the

SELF∩VALUE mask). The additivity of the effect in this region is illustrated by the common of responses to winning (WIN>noWIN: -2,48,-4, $t(114)=7.60$; $Z=6.82$, $k=193$; $p<.05$, FWE-corrected) as well as control (HC>MC; -8,48,-12, $t(114)=6.13$; $Z=5.69$, $k=162$, $p<.05$, small-volume FWE-corrected). The localization of this effect thus mirrors how the processing of value and self-reference converges in the vmPFC53,60. Aside from these rather additive effects of success and internal control beliefs, one might hypothesize that vmPFC activity responds in a super-additive fashion or spreading activation to the convergence of these factors. That is, vmPFC activity might differ more strongly between WIN and noWIN for HC than for MC outcomes. We thus tested this interaction ($[HC:WIN>HCnoWIN]>[MC:WIN>MCnoWIN]$), which did not yield any significant effects in any of the masks (even at more lenient thresholds at $p<.001$, uncorrected), nor on the whole-brain level.”

Methods:

II. 873-877: “For the analyses at the second level, we focused on the outcome phase of the task by implementing a within-subjects ANOVA including the individual first-level contrast images for MC:WIN, MC:noWIN, HC:WIN, and HC:noWIN. Using this approach we computed main effects of condition (HC>MC) and outcome valence (WIN>noWIN), the conjunction $(HC:WIN>HC:noWIN) \cap (HC:WIN>MC:WIN)$, as well as the interaction $((HC:WIN>HC:noWIN) > (MC:WIN>MC:noWIN))$.”

Figure 5:

Figure 5. Additive effect of winning under high control and parametric effects of emotion ratings. **a** Activations show the specific effect of HC:WIN (i.e. 2-way conjunction $[HC:WIN>HC:noWIN] \cap [HC:WIN>MC:WIN]$) displayed at $p<.005$, uncorrected. The conjunction contrast survived small-volume FWE-correction at $p<.05$ inside the SELF∩VALUE mask in vmPFC. Bar plot represents differences of neural activation in vmPFC between HC:WIN and the other three outcomes. **b** Parametric modulation effect of happiness (red-yellow) and pride ratings (blue-green) at outcome presentation, displayed at $p<.001$, uncorrected for illustrative purposes. Bar plot shows group-level parameter estimates of the parametric modulations at the peak of each parametric effect inside the SELF∩VALUE mask. Violet outlines show SELF∩VALUE mask. Error bars are +/- 1 standard error of the mean.

Table S19-S20:

Table S19. 2-way conjunction small-volume-corrected inside SELF∩VALUE, critical T = 3.164

set		cluster				peak			MNI coordinates (mm)				
p	c	p(FWE)	p(FDR)	k	p(unc)	p(FWE)	p(FDR)	T	Z	p(unc)	x	y	z
.050	1	.025	.486	12	.486	.022	.435	3.45	3.36	<.001	-4	48	-10

Note. Effects listed are for the contrast $[HC:WIN>HC:noWIN] \cap [HC:WIN>MC:WIN]$

Table S20. 2-way conjunction inside SELF∩VALUE, $p<.005$, uncorrected

set		cluster				peak			MNI coordinates (mm)				
p	c	p(FWE)	p(FDR)	k	p(unc)	p(FWE)	p(FDR)	T	Z	p(unc)	x	y	z
1	2	.986	.911	80	.178	.984	.691	3.45	3.36	<.001	-4	48	-10
		1.000	.911	1	.911	1.000	.692	3.06	3.00	.001	0	42	4

Note. Effects listed are for the contrast $[HC:WIN>HC:noWIN] \cap [HC:WIN>MC:WIN]$

b. Second, the PPI results are interesting. The main result involves increased connectivity in VS and VMPFC as a function of receipt of task outcomes in the HC. Was this all outcomes? Or a contrast of win vs no win? Basically, is it driven by win or no win?

c. Third, also on the PPI, it would be useful to perform the same PPI with MC trials to see if the VS-MPFC connectivity is also there.

Thank you very much for these comments and please apologize if we haven't been clear enough in the earlier version. We would like to respond to the points 5b and 5c collectively, because we think that this allows clarifying a misunderstanding that might have been related to lacking precision in the wording of the original draft. Specifically, the description of our results regarding the psychophysiological interaction (PPI) analysis was misleading, suggesting that we performed the PPI analysis in the HC condition only, while we truly ran a contrast of connectivity differences in HC > MC. We have now clarified the initial sentence regarding the PPI results in order to avoid such a misunderstanding when reading the manuscript (see ll. 287 and 298 in the results section of study 2). We would furthermore like to elaborate on the rationale of our original analysis and have also performed new analyses suggested by the reviewer (regarding comment 5b).

The original PPI analysis contrasts VS connectivity for both WIN and noWIN outcomes between HC and MC. Thus, this analysis shows that the activity time-course of the left VS that tracks the valence of outcomes (WIN>noWIN) is more strongly correlated with the activity of cortical midline structures (CMS) when outcomes are received under HC than when they are received under MC. Importantly, the contrast which defined the region from which to extract the time-course of activity (i.e. WIN>noWIN) was orthogonal to the contrast used for defining the PPI term (HC>MC). This is important for methodological reasons as orthogonal contrast for selecting time-courses and contrasts in the PPI improve the statistical power of the analyses and it is generally recommended to build experimental designs so that they result in orthogonal contrasts of interests. As WIN and noWIN varies within HC and MC with equal frequency our design ensures that the time-courses extracted from regions that show an effect for success (i.e. the VS) are independent to the factor of control.

Yet, as suggested by the reviewer, it is in fact interesting to see whether the PPI effect we report so far was mainly driven by WIN or noWIN outcomes (comment 5b), and whether we can identify connectivity effects for the HC and MC conditions separately (i.e. without contrasting the two, comment 5c). Hence, in order to answer this question, we ran a series of additional PPI analyses.

For the first analysis, we constructed a new PPI regressors in SPM using the interaction of the left VS time-course and the contrast HC:WIN>MC:WIN. This analysis tests whether functional connectivity of left VS is stronger for HC:WIN outcomes than for MC:WIN outcomes, thus excluding noWIN outcomes. Equivalently, we constructed a second PPI term based on the left VS time-course and the contrast HC:noWIN>MC:noWIN, thus excluding WIN outcomes. We included these two PPI terms in a new first-level GLM analyses of each subject, which otherwise included the left VS time-course, the two contrasts used for construction of the PPI terms, and all regressors that were also included in the original PPI reported in the draft (i.e. two regressors modeling task phases, two regressors modeling rating phases, as well as six realignment parameters and their first derivatives). For each subject, we then contrasted the PPI effect for WIN outcomes against the PPI effect for noWIN outcomes and included these first-level contrast images in a second-level one-sample t-test.

The second set of analyses focused on connectivity during HC and MC outcomes separately. For each of these analyses, we constructed a PPI term based on the interaction of the left VS time-course with the respective contrast (i.e. HC>0 and MC>0). For each subject, two separate first-level GLM analyses were run, which included as predictors the respective PPI term, the left VS time-course, and the contrast used for constructing the PPI term. Additionally, we added regressors for the two outcomes not included in the PPI (e.g. for the HC PPI, two regressors modelling MC:WIN and MC:noWIN outcomes were included, and vice versa). Last, in each GLM, we added all remaining regressors that were also included in the original PPI analysis (i.e. two regressors for modelling the task phases, two regressors modelling the rating phases, as well as six realignment parameters and their first derivatives). Finally, for both the HC PPI as well as the MC PPI, we ran separate second-level one-sample t-tests on the first-level contrast images of the respective PPI term.

We included descriptions of these additional PPI analyses in the methods section and report the results of these analyses.

Results:

II. 299-305: “In a set of control analyses we tested whether dynamics in functional connectivity were driven by WIN or noWIN outcomes. Comparing the PPIs contrasting HC:WIN>MC:WIN and HC:noWIN>MC:noWIN did not reveal significant differences in the connectivity dynamics in any of the a-priori masks nor on the level of the whole brain. Further analyses additionally showed that the HC>MC PPI effects were driven by increased functional connectivity of left VS for HC outcomes (see supplementary tables S34 & S35, and supplementary figure S2).

Methods:

II. 912-929: “In a first follow-up analysis, we focused on whether the PPI described above was driven by WIN or noWIN outcomes. Therefore, we constructed two PPI terms for each subject by computing the interaction of the left VS time-course with the contrasts HC:WIN>MC:WIN and HC:noWIN>MC:noWIN, respectively. Aside from these two PPI terms, we included regressors modelling the two task phases, two rating phases, the two contrasts used for constructing the PPI terms, the activation time-course of the left VS, as well as six realignment parameters and their first derivatives. The first level images contrasting the WIN and noWIN PPI terms were then taken to a second-level one-sample t-test, to test whether there were significant connectivity differences for WIN and noWIN outcomes.

A second set of follow-up analyses tested for functional connectivity of the left VS only during HC and only during MC outcomes. For each subject, we computed the interaction of the left VS activation time-course with the contrast HC>0 and the contrast MC>0, respectively. We then set up two GLMs for each subject, including, aside from the respective PPI terms, regressors modelling the two task phases, two rating phases, the contrast used for constructing the PPI term, the activation time-course of the left VS, as well as six realignment parameters and their first derivatives. In addition, for the PPI modelling HC connectivity, the outcome phases for MC (MC:WIN, MC:noWIN) were included as two additional regressors. Vice versa, for the PPI modelling MC connectivity, the outcome phases for HC (HC:WIN, HC:noWIN) were included as two additional regressors. We then computed two second-level one-sample t-tests on the first level PPI effects (contrast images for HC PPI and MC PPI).”

Supplementary figure S2:

Figure S2. Functional connectivity of the left ventral striatum (VS). **a** For HC outcomes, the PPI term was based on the interaction of the left VS time-course and the contrast HC>0. Only positive connectivity effects were found. **b** For MC outcomes, the PPI term was based on the interaction of the left VS time-course and the contrast MC>0. Connectivity effects for this PPI were generally weak, so the effects are displayed on very liberal thresholds ($p < .05$, uncorrected) for illustration purposes. Blue-green indicates negative connectivity effects, red-yellow indicates positive connectivity effects.

Supplementary tables S34-S35:

Table S34. Left VS PPI (HC>0) wholebrain $p < .001$, uncorrected, $k \geq 50$

set		cluster				peak					MNI coordinates (mm)		
p	c	$p(\text{FWE})$	$p(\text{FDR})$	k	$p(\text{unc})$	$p(\text{FWE})$	$p(\text{FDR})$	T	Z	$p(\text{unc})$	x	y	z
<.001	10	<.001	<.001	2973	<.001	.028	.276	5.75	4.83	<.001	-10	26	60
						.033	.276	5.68	4.79	<.001	-8	56	38
						.297	.499	4.76	4.18	<.001	4	38	26
		.134	.107	154	.017	.256	.499	4.83	4.23	<.001	-40	8	56
						.532	.527	4.44	3.95	<.001	-42	16	52
						.819	.648	4.09	3.69	<.001	-32	14	50
		.028	.028	254	.003	.262	.499	4.82	4.22	<.001	48	-64	42
						.845	.675	4.06	3.66	<.001	52	-56	34
						.970	.740	3.77	3.44	<.001	46	-60	30
		.002	.003	462	<.001	.277	.499	4.79	4.20	<.001	42	28	-6
						.379	.499	4.63	4.09	<.001	46	36	-12
						.419	.499	4.58	4.05	<.001	34	22	-16
		<.001	.001	572	<.001	.293	.499	4.76	4.18	<.001	-40	40	-6
						.385	.499	4.63	4.08	<.001	-38	34	2
						.567	.544	4.40	3.92	<.001	-44	42	-16
		.124	.107	159	.015	.343	.499	4.69	4.13	<.001	-46	-68	40
		.002	.003	466	<.001	.444	.499	4.55	4.03	<.001	-4	-4	6
						.473	.499	4.51	4.00	<.001	-10	14	6
						.596	.544	4.37	3.90	<.001	-14	4	12
		.013	.015	310	.001	.481	.499	4.50	3.99	<.001	8	-56	34
						.882	.691	3.99	3.62	<.001	0	-56	36
						.920	.702	3.92	3.56	<.001	-4	-70	56
		.657	.636	56	.125	.683	.550	4.27	3.82	<.001	-28	20	-20
		.465	.413	79	.073	.914	.702	3.93	3.57	<.001	40	-60	-26
						.976	.764	3.74	3.42	<.001	30	-66	-24

Table S35. Left VS PPI (MC>0) wholebrain $p < .001$, uncorrected

set		cluster				peak					MNI coordinates (mm)		
p	c	$p(\text{FWE})$	$p(\text{FDR})$	k	$p(\text{unc})$	$p(\text{FWE})$	$p(\text{FDR})$	T	Z	$p(\text{unc})$	x	y	z
1	3	1.000	.858	1	.858	.992	.996	3.67	3.37	<.001	-20	-30	10
		.999	.858	2	.784	1.000	.996	3.37	3.13	<.001	-40	-86	14
		1.000	.858	1	.858	1.000	.996	3.33	3.09	<.001	-26	-88	18

Comment #6: For the fMRI design, were ratings evenly distributed across wins and no wins (and conditions?) That is, would HC pride trials randomly follow more wins than no win trials for example? Or was the order of subjective ratings also sampled in a way that ensured even distribution across outcomes and conditions?

A total of 48 ratings for pride (24) and happiness (24) was evenly distributed across all combinations of HC/MC and WIN/noWIN, with 6 ratings of either pride or happiness for each of the 4 possible trial types. The assignment of ratings to trials was predefined so that ratings would in fact be evenly distributed across the experiment and additionally, the assignment of pride and happiness ratings to specific trials was counterbalanced across participants. That is, 19 of the 39 participants rated pride for those trials on which the other 20 participants rated happiness, and vice versa.

Information on the nature and distribution of ratings across conditions can be found on pp. 31-32 of the original submission (lines 709 to 720 of the revised manuscript). In order to make this information more accessible to the reader, we have also included more details on this in the caption of figure 1.

Caption figure 1:

“Overview of the three studies. The task was presented in each study with varying numbers of trials, conditions, and ratings. In study 1, both pride and happiness were rated after each outcome. In study 2, either pride and happiness were rated after a given outcome. Ratings were evenly distributed across the four possible outcomes, and the order of pride and happiness ratings was predefined and counterbalanced across subjects. In study 3, only pride was rated, and predefined ratings were evenly distributed across all possible outcomes. See methods for details.”

Comment #7: Are happiness and pride correlated in the experiments?

This is a very good comment indeed, thank you very much for bringing this up. The information regarding the correlation of pride and happiness was in fact missing in the original submission. The revised submission now reports these analyses in the results section of study 1 (lines 186-187) and more detailed information can be found in the supplementary materials, for both studies 1 and 2 (tables S10, S11, & S12). In study 3, only pride was rated, so no correlations with happiness could be computed. While focusing on the correlation, your question also directs to a comment of reviewer #2 (comment #1) referring to the implications of a potential association between both affective concepts. Please also see our response to their comment #1 below which addresses related implications and conceptual issues.

Since pride and happiness ratings were differently distributed in study 1 and study 2 we ran different analyses to disentangle the association of both affective concepts. First, only in study 1 we could test whether pride and happiness ratings are correlated on the within-subject level, as this was the only study in which subjects rated both emotions for the same outcomes. The mean within-subject correlation of pride and happiness across all conditions is $M=.671$ ($SD=.209$). Thus, ratings of these emotions are correlated in our task, which is to be expected given that both are strongly characterized by positive affective experience and correlations vary across individuals.

Second, in addition to the within-subject correlations reported above, we computed the mean pride and happiness ratings of each individual subject for each condition in both studies 1 and

2. In study 1, correlations of pride and happiness between subjects ranged from $r = .68$ (LC:noWIN) to $r = .82$ (HC:WIN). In study 2, the lowest (absolute) correlation between pride and happiness across subjects was $r = .52$ (MC:WIN), and the highest was $r = .95$ (HC:noWIN). Thus, subjects who reported higher intensities of happiness generally also reported higher levels of pride in the same condition. A detailed table including all these correlations of pride and happiness between subjects is now included in the supplementary material (see supplementary tables S10, S11, & S12).

Last, we tested whether the pride and happiness responses were correlated (i.e. the term [HC:WIN-HC:noWIN] – [MC:WIN-MC:noWIN] computed separately for pride and happiness ratings for each subject). In both studies 1 and 2, medium to large correlations of pride and happiness responses were found (study 1: $r = .38$, one-sided $p = .008$; study 2: $r = .47$, one-sided $p = .001$). This indicates that subjects whose pride was more strongly modulated by HC as compared to MC outcomes, also reported a stronger control-dependent modulation of happiness. Nevertheless, we note that the pride response was significantly larger than the happiness response in both studies 1 and 2, as indicated by the 3-way interactions in the repeated measures ANOVAs. We have referred to the correlation between pride and happiness responses in the results sections of studies 1 & 2.

Results:

II. 185-189: “Besides these effects on positive affect and the generally strong association between pride and happiness ratings (within-subject Pearson’s r : $M = .67$, $SD = .21$, $t(39) = 20.29$, one-sided $p < .001$; for between-subject correlations see supplementary tables S10 and S12), the pride response (i.e. the difference [HC:WIN-HC:noWIN]-[MC:WIN-MC:noWIN]) increased more strongly from MC to HC than the happiness response ($t(39) = 3.69$, one-sided $p < .001$).”

II. 209-215: “Subjects experienced much higher internal control in HC than in MC (paired t -test: $t(38) = 9.56$, $p < .001$, $d = 1.53$, 90% $CI = [1.13; 1.91]$), and affect ratings differed between tasks with different levels of control (rmANOVA: main effect of condition: $F(1,38) = 8.38$, $p = .006$, $\eta_p^2 = .18$), outcome valence ($F(1,38) = 174.83$, $p < .001$, $\eta_p^2 = .82$) and their interaction ($F(1,38) = 40.23$, $p < .001$, $\eta_p^2 = .51$) supporting the impact of successes and internal control beliefs on positive affect (see also supplementary tables S11 & S12 for the association of pride and happiness).

Supplemental tables 10-12:

Table S10. Pearson correlations of pride and happiness ratings across subjects (study 1)

		proud						happy						
		LC		MC		HC		LC		MC		HC		
		no WIN	WIN	no WIN	WIN	no WIN	WIN	no WIN	WIN	no WIN	WIN	no WIN	WIN	
proud	LC	WIN	.73											
	MC	noWIN	.98	.72										
		WIN	.72	.97	.69									
	HC	noWIN	.96	.69	.97	.68								
		WIN	.62	.83	.57	.90	.55							
	LC	noWIN	.68	.42	.70	.36	.70	.18						
WIN		.50	.73	.46	.79	.45	.80	.41						
happy	MC	noWIN	.67	.42	.69	.36	.71	.19	.99	.43				
		WIN	.47	.69	.42	.78	.41	.80	.36	.98	.37			
	HC	noWIN	.62	.39	.64	.32	.69	.14	.96	.41	.97	.35		
		WIN	.46	.67	.41	.74	.39	.82	.34	.95	.36	.96	.30	

Table S11.
Pearson correlations of pride and happiness ratings across subjects (study 2)

		proud				happy			
		MC	MC	HC	HC	MC	MC	HC	HC
		no WIN	WIN	no WIN	WIN	no WIN	WIN	no WIN	WIN
proud	MC	WIN	.24						
	HC	noWIN	.88	.19					
		WIN	.05	.65	-.08				
	happy	MC	noWIN	.89	.08	.91	-.17		
WIN			-.17	.52	-.12	.66	-.28		
HC		noWIN	.90	.22	.95	-.09	.90	-.17	
		WIN	-.34	.45	-.37	.75	-.48	.86	-.36

Table S12.
Spearman correlations of pride and happiness responses across subjects (studies 1 & 2)

	Spearman's rho	90% CI		one-sided p
		lower	upper	
Study 1	.42	.18	.62	.003
Study 2	.24	-.03	.48	.067

Comment #8: For Exp.3 (and to a degree the vmPFC result in 2), it would be useful to discuss Wang et al., 2019, Cerebral Cortex, which also showed that individuals were willing to give up value in order to attain control.

We thank the reviewer for this suggestion and the link of these important results to our study. We fully agree that the article by Wang et al. nicely relates to the questions targeted by our work. In fact, it provides further evidence regarding the value of control and its neural processing in the vmPFC. This study employed a clever choice paradigm comparable to the one we used in study 3 of the present manuscript. Precisely, participants were presented with different monetary incentives for choosing or not to exert control, and the results show that individuals with a stronger preference for control were willing to give up money in order to exert control themselves instead of ceding it to the computer.

These findings are close to ours, but with respect to the reviewer's first comment and our response to it, we note that in our view, the two studies focus on different aspects of perceived behavioral control. Specifically, Wang and Delgado convincingly demonstrate that subjects are willing to give up money in order to have any control at all, that is, in order to make choices themselves. Our study extends these findings and shows a preference for informed choices that allow self-evaluation (as detailed above) versus choices that do not allow such self-evaluation and relate these to affective dynamics during task execution. Hence, while Wang and Delgado (2019) focus on free choice (as compared to no choice) as a basic necessity for exerting control, our study highlights how individuals strive for situations in which they know why to behave in a certain way as compared to choosing randomly between two options.

Furthermore, the study by Wang and Delgado (2019) also shows stronger engagement of the vmPFC in subjects with a more pronounced control preference when choosing between exerting control themselves or letting the computer exert control for them. This finding nicely extends the literature on value computation, control preferences, and functionality of the vmPFC (Kable & Glimcher, 2007; Leotti & Delgado, 2011; Leotti et al., 2010; Maier & Seligman, 2016). We would expect that the vmPFC is also relevant when individuals choose between

tasks akin to our HC and MC conditions, but the data from the present manuscript do not allow to test this hypothesis.

We have included a discussion of these findings and reference to the article by Wang and Delgado in the results, as well as in the discussion section.

Results:

II. 388-391: "In order to further embed our data into the existing literature on perceived control and task preferences (Leotti et al., 2010; Wang & Delgado, 2019), we also tested the association between the three indices of control preference and individual differences in experienced control over the different tasks."

II. 398-401: "Collectively, these findings support and extend previous results that showed that mere choice is preferable to not having choices (Leotti & Delgado, 2011; Wang & Delgado, 2019) by highlighting the notion of instrumental control over the outcomes for building preferences (Ly et al., 2019), and emphasizing that self-related positive affect might be a relevant factor for mediating motivated behavior (Williams & DeSteno, 2008)."

Discussion:

II. 504-509: "In fact, a recent study by Wang & Delgado (2019) provides first evidence supporting this notion. The authors show that individuals with a stronger preference for control show higher vmPFC activity when choosing between exerting control themselves or ceding control to the computer. In that sense it seems a plausible next step to directly assess how vmPFC activity during outcome valuation relates to behavioral preferences when outcomes are under internal control and how the connectivity dynamics of the VS can inform motivated behavior."

II. 542-547: "Interestingly, previous studies have conceptually linked the vmPFC to illusions of control, distorted probability estimates, and positive affect (Kool, Getz, & Botvinick, 2013), and there is evidence relating activity in this region to behavioral preferences for subjectively controllable tasks (Wang & Delgado, 2019). In concert with these findings, the present results provide a hint for clinical research to refocus on perturbed beliefs of control over the environment to understand affective and motivational symptoms as well as altered neural dynamics associated with relevant psychiatric conditions."

Reviewer #2 (Remarks to the Author):

This manuscript reports 3 experiments that evaluate the relationship between affective reactions to outcomes associated with varying levels of control. A novel choice task was used to manipulate control (1. no choice/control (LC), 2. binary choice/gambles (MC), and 3. perceptual discrimination task that is ostensibly ability-based (HC)) and pride and happiness ratings are evaluated after outcomes (Win vs. No-Win). Manipulation checks show that subjective ratings of control increase with the levels of the task manipulation. In Study 1, happiness and pride increased following wins than no wins, but more so as controllability increased, and more so for pride than happiness. In Study 2, value-based regions of the brain (vMPFC, striatum) responded more strongly to wins than no-wins, self-processing regions (vMPFC, ACC, precuneus) responded more strongly to HC>MC conditions, and vMPFC related to both value and self-processing responded more strongly to wins in high control conditions and to pride more so than happiness. Moreover, the vMPFC region showed higher covariation with striatum in the HC condition. Finally, in Study 3, individuals were willing to forgo monetary reward to select HC choices over MC choices, and their propensity to choose HC over MC (when their value was equivalent) correlated to pride responses.

The task is clever and well-designed to manipulate perceived control, and the topic is timely and interesting. It is also commendable that the authors pursued their questions across a few experiments with independent samples. However, I have a number of concerns that limit the interpretation and impact of the results, which I outline below:

We thank the reviewer for the positive and constructive feedback and the appreciation of the study design. We think that we have addressed all your comments and included your recommended analyses in our results. Your suggestions and remarks helped a lot for revising the manuscript. Please find a point-by-point answer to your comments below.

Comment #1a: Given that pride and happiness and importantly their difference is so critical to the current interpretation, it would be important to evaluate the extent to which pride and happiness judgments are correlated. Based on the nature of the task – making perceptual discrimination judgments about luminance and receiving manipulated feedback – it could be that people are experiencing pride and happiness, or some mildly positive response that straddles the two. If people are asked to assign a number to pride and to happiness, they'll do so, but it is unclear if the experience itself represents pride and happiness and whether these are separable.

Your comment addresses an important empirical and conceptual issue and relates to a question reviewer #1 has asked with respect to the correlation of pride and happiness (please see reviewer #1, comment #7). Besides the empirical investigation of the association between both affective concepts in our study, which we have addressed in greater detail above and briefly summarize below, we here aim to elaborate more thoroughly for why believe it is important to examine both constructs concurrently. To do so, we broke your comment down into two sections, a first rather general part and a second, which we think more specifically illustrates your point.

Variation in the pride and happiness ratings in response to the here presented outcomes are associated both on the within-subject as well as on the between subject level. The effect sizes suggest a strong association with ~40% of the variance shared on average on the within subject level as well as between ~25% and ~80% on the between subject level, depending on conditions and study. These associations are important to consider but not surprising at all. With respect to theory of self-conscious emotions, happiness could in fact be construed rather independent of pride while the experience of pride is closely entangled with a concurrent happiness response. The motivation for including happiness ratings in the current study was

thus not to show that both concepts are unrelated but to verify that we in fact measure differences in the qualitative construal of affect and not merely rely our interpretation on an increased positive valence of positive outcomes under high control. This can be rooted in a constructivist theory of emotion, which proposes that emotions are not naturally existing entities, but rather abstract concepts that allow to categorize and create meaning out of internal states (like increased arousal, hedonic tone, etc.; Lindquist & Barrett, 2008; Roy, Shohamy, & Wager, 2012; Seth, 2013). Importantly, it is assumed that the context in which certain internal states occur will shape which emotional concepts are likely to be applied. With regard to the present scenario, obtaining WIN or noWIN outcomes leads to shifts in hedonic state with WIN outcomes having much greater positive valence. If the task context then allows attributing these changes in hedonic state to own actions, the probability with which an individual conceptualizes her internal states as pride (or reductions thereof) increases. In contrast, if one does not feel responsible for the event causing a given state, references to pride should decrease in theory. Thus, receiving positive information increases happiness (Rutledge, Skandali, Dayan, & Dolan, 2014), and relating this information to own action additionally leads to pride, as shown in the present manuscript. In this line, according to theory of self-conscious emotions it is less likely to experience pride without experiencing happiness than vice versa (Tracy & Robins, 2004). Therefore, we expected ratings of these emotions to be substantially correlated and it would be conceptually more problematic if both emotions were not correlated at all, as this would question the validity of our measures and the underlying theory. Despite the relatively strong correlation of the two emotions, however, and within the space to construe the emotions on the given scales of pride and happiness, we think that participants' responses well reflect the qualitative shift in the construal of affective experiences if internal control beliefs increase.

We have now included a brief clarification of our rationale in the introduction, and refer to the new analyses in the results section. Also, we have now aimed to improve the clarity of our argument throughout the manuscript and explain how internal control beliefs should relate to changes positive affect in general and a qualitative shift towards pride more specifically. Due to the various minor changes we only highlight the most prominent changes with regards to your comment here.

Introduction:

II. 81-85: "Theories on self-conscious affect (Tangney, Stuewig, & Mashek, 2007) and appraisal theories (Roseman, Antoniou, & Jose, 1996; Scherer, 2001) predict that control beliefs shift the valuation of outcomes and result in distinct patterns of affective experience. While internal control beliefs should relate to dynamics in positive affect due to their intrinsic value, the attribution of outcomes to controllable and internal causes is also a necessary condition for the experience of the self-conscious affect of pride (Tracy & Robins, 2004)."

II. 101-102: "As predicted by theory, positive affect and pride dynamics in particular should be more strongly modulated by outcomes if these are perceived to depend on one's own behavior."

II. 127-129: "More precisely, we hypothesize successes and greater control beliefs to increase positive affect since both carry intrinsic value. Moreover, we expect to observe a qualitative shift in the affective report with stronger references to pride in context of internal control beliefs."

Results:

II. 182-189: “The main effects of outcome valence ($F(1.00,39.00)=60.92$, $p<.001$, $\eta_p^2=.61$) and condition ($F(1.31,50.93)=60.92$, $p<.001$, $\eta_p^2=.43$), as well as their two-way interaction ($F(1.37,53.55)=48.51$, $p<.001$, $\eta_p^2=.55$) additionally show how internal control beliefs drive positive affect in response to stimulus outcomes in general. Besides these effects on positive affect and the generally strong association between pride and happiness ratings (within-subject Pearson’s r : $M= .67$, $SD=.21$, $t(39)=20.29$, one-sided $p<.001$; for between-subject correlations see supplementary tables S10 and S12), the pride response (i.e. the difference [HC:WIN-HC:noWIN]-[MC:WIN-MC:noWIN]) increased more strongly from MC to HC than the happiness response ($t(39)=3.69$, one-sided $p<.001$).”

II. 193-198: “If such self-attribution is possible, internal control increases beyond having a choice or not, effectively increasing the self-relevance of outcomes and changing their affective valuation by driving positive affect and qualitative shifts in the construal of the experience towards pride (Tracy & Robins, 2004). In contrast, merely having a choice may be rewarding in itself (Leotti & Delgado, 2011), but falls short of internal control beliefs as defined in classic theories (Ajzen, 2002; Bandura, 1977; Rotter, 1966) and should have less impact on this self-evaluative aspects of positive affect (Roseman et al., 1996; Tangney et al., 2007; Tracy & Robins, 2004).”

II. 209-221: “Subjects experienced much higher internal control in HC than in MC (paired t-test: $t(38)=9.56$, $p<.001$, $d=1.53$, 90% CI=[1.13; 1.91]), and affect ratings differed between tasks according to the level control (rmANOVA: main effect of condition: $F(1,38)=8.38$, $p=.006$, $\eta_p^2=.18$), outcome valence ($F(1,38)=174.83$, $p<.001$, $\eta_p^2=.82$) and their interaction ($F(1,38)=40.23$, $p<.001$, $\eta_p^2=.51$) supporting the impact of successes and internal control beliefs on positive affect (see also supplementary tables S11 & S12 for the association of pride and happiness). Comparably to study 1, however, dynamics in pride in response to task outcomes differed from those for happiness in dependence of control beliefs (condition*outcome valence*affect: $F(1,38)=9.08$, $p=.005$, $\eta_p^2=.19$).[...] The resulting pride response was significantly larger than the happiness response ($t(38)=3.01$, one-sided $p=.005$), emphasizing how the construal of self-related positive affect hinges on the subjective belief that outcomes depend on one’s own actions (for full rmANOVA effects, see supplementary table S2).”

Discussion:

II. 427-445: “Overall, our findings provide evidence for the link between internal control beliefs, positive affect and the experience of pride in particular. This concords with psychological theories of self-conscious affect (Tracy & Robins, 2004) and appraisal theories (Roseman et al., 1996; Scherer, 2001), stating that internal control allows attributing task outcomes to internal causes, such as one’s abilities. Self-evaluative processes are crucial to the understanding of self-conscious affect like pride, differentiating them from other constructs that people use to describe their affect, such as happiness. It should be noted however, that positive affect was generally increased in response to successes under high control and both affective experiences were highly correlated with each other across the different tasks. This is not surprising as happiness and pride both highly relate to a core dimension of positive affect and pleasantness (Roseman et al., 1996). In theory, it is thus very unlikely that people report feelings of pride without also referring to happiness (Tracy & Robins, 2004) which was also

reflected in our data as the affective report for happiness exceeded the reported experience of pride also in the highly controllable conditions. Since outcomes obtained under internal control beliefs are informative about one's abilities (Müller-Pinzler et al., 2019), participants can use these outcomes to infer to what extent they succeed at actually exerting control. Regarding their affective report, this was reflected by a qualitative shift in the response, i.e. the fact that the concept of pride was more readily referred to in the task allowing higher levels of internal control. We did not observe such construal of affect for uncontrollable events, where increased happiness, but relatively lower pride was observed in response to positive outcomes. This supports the theoretical assumption that control beliefs are not merely related to increased positive valence but also contribute to a qualitatively distinct construal of affect when outcomes are obtained (Roseman et al., 1996; Tracy & Robins, 2004).

Comment #1b: For instance, the difference in pride between Win and No Win is more influenced by the control manipulation (HC > MC) than happiness, which could lead to the interpretation that "the resulting pride response was significantly larger than the happiness response". But it is the difference of differences that differs by the control manipulation, and not the overall degree of pride or happiness being experienced. The levels of happiness seem larger overall than levels of pride, and the difference between Win and NoWin in HC condition in happiness seems equivalent (Study 2) if not larger (Study 1) than that for pride.

Thank you very much for bringing up this argument in greater detail. It is of course true that the overall intensity of happiness is higher than that of reported pride, but in our view, this does not reduce the validity of our findings, interpretations, and implications.

Regarding the intensity of affect, we did not expect pride ratings to be as high or even higher than happiness ratings. This is due to the assumption that visual discrimination abilities as in our task (HC condition) might not be of high self-relevance to the majority of our subjects, which is however central to the experience of strong self-conscious affect. In general, the necessity of self-relevance of a task for the experience of self-conscious emotions makes it difficult to create experimental designs that elicit strong levels of pride in the majority of subjects (Tracy & Robins, 2004). It will be even more difficult in case one aims to design experiments that allow for the controlled presentation of repeated events in rather short periods of time and can also be conducted in the restricted environment of neuroscience laboratories. As a consequence, the vast majority of studies examining pride and other self-conscious emotions in the neurosciences relied on presenting standardized vignettes and using imagination techniques which are questionable with regards to inducing genuine and strong affective experiences for their own reasons (see however Müller-Pinzler et al., 2015 for an exception). In this line, our study is one of the few examples that aims to create experimental conditions that should be able to induce the experiences of pride, even though they might be comparably small and do not reflect the full-blown affective experience of pride that people report in everyday life and when mastering a highly self-relevant task.

However, our goal was not to induce the most intense feelings of pride, but rather to show how the unique pattern in pride and happiness emerges in response to outcomes and how this depends on internal control beliefs. That is, the reason why we included happiness ratings in our tasks was to demonstrate that control-dependent affective responses are most prominent in reported pride, and not simply attributable to changes in general positive affect. This renders the overall intensity of the reported emotions less relevant. To highlight this point, we also computed the difference in pride between HC:WIN and MC:WIN outcomes, and compared this to the equivalent term for happiness. In both studies 1 and 2, this increase was stronger for

pride than for happiness ratings (study 1: $t(39)=4.039$, one-sided $p<.001$; study 2: $t(38)=2.822$, one-sided $p=.004$), while the overall intensity of happiness was of course higher in all four conditions in both study 1 and study 2 (see figures 1 and 2 in the manuscript).

We have added these analyses in the result section.

Results:

II. 177-181: "To highlight that internal control over successes impacts pride more strongly than happiness, we also computed the difference in pride between HC:WIN and MC:WIN outcomes specifically, and compared this to the equivalent term for happiness. This increase was stronger for pride than for happiness ratings ($t(39)=4.039$, one-sided $p<.001$), while the overall intensity of happiness was higher in all four conditions (see figure 1)."

II. 221-223: "Emphasizing the effect of internal control on affective responses, the difference in pride between HC:WIN and MC:WIN outcomes was significantly larger than the equivalent term for happiness ($t(38)=2.82$, one-sided $p=.004$), as was the case in study 1."

Comment #2: For the HC condition, if feedback was pre-determined, then the actual outcomes are still chance based. If pride depends upon feelings of mastering a task, then are people really feeling pride if the outcomes do not depend on mastery or ability? Did participants suspect that feedback was not tied to performance, as their performance on HC trials could not improve over time? This sense of improvement seems important for mastery and hence pride.

We want to structure our reply to this comment in two sections and start with the second question whether participants suspected that feedback was not tied to performance. As we have stated in the manuscript, we excluded those subjects who did not believe our cover-story. In order to identify those subjects doubting the truthfulness of the feedback, the post-experimental interview included written questions asking subjects a) whether they had any idea about the general purpose of the study; b) whether they noted any peculiarities during the study; and c) whether they wanted to note anything else about the experiment. After subjects had a chance to answer these questions, the experimenter debriefed each subject, by first asking broadly about anything the subject would like to note about the task and then asking more specifically about the separate conditions. After informing the subjects that feedback was manipulated they were also asked whether they had suspected this. Answers to these questions were the criteria to exclude a given subject because the cover story did not work sufficiently well. In total, 12.3% of subjects reported some sort of skepticism and were excluded (i.e. 19 out of 154 tested subjects, see table 1). Importantly, this does not imply that subjects felt they were unable to master the task or did not feel to improve over time. In fact, none of these subjects was skeptical because of this issue. Rather, on some trials these subjects indicated that they were very certain about their identification of the brightest square, while feedback was negative and thus inflicted doubt about whether feedback was truthful and linked to performance.

Your first question appears to be conceptually related to your previous comment on whether this paradigm is able to induce strong pride experience in the individuals. As we have stated above, our aim was not to induce strong pride experiences per se but stimulate the construal of different affective qualities related to internal control beliefs. Further, we want to elaborate what the concept of mastery contributes to the understanding of pride and whether your comment is critical for the implications and conclusions that we can draw from our study.

While we are in agreement with the notion that mastery (indicated by improvement over time) can drive pride, we would not posit that it is a necessary condition for experiencing pride. For instance, it is likely that individuals who try to get better at a specific task and who monitor their improvement over time will be proud in case this intrapersonal performance comparison over time indicates that they have, in fact, become better. However, another situation in which individuals would likely experience pride is an interpersonal comparison, such as beating an opponent in a game of chess. In this situation, the improvement of one's abilities over time does not necessarily constitute the central phenomenological experience that drives emotional responses at that moment. Yet, importantly, we argue that in both of these situations, i.e. the intra- and interpersonal comparisons, it is relevant that the observed outcomes are attributable to one's own efforts or abilities, and not to chance or other external reasons. In fact, attributing events to own actions is a necessary prerequisite for experiencing mastery or improvement over time, as events that are perceived to be due to chance are per se not informative about one's ability. This does, however, not mean that pride is only elicited when one experiences improvement over time, or mastery.

Another argument regarding the point that outcomes are chance based by design and thus imply the lack of experience of mastery is that the included subjects did not only rate the HC task to be more controllable than the other tasks, but in study 3 also rated the HC task as more difficult than the MC task (see table S9; difficulty was not rated in studies 1 & 2). In our view, even if a task is controllable, high difficulty can impede high success rates, which allows for chance-level (i.e. 50/50) outcomes to occur. Thus, receiving 50% WINS and 50% noWINS does not necessarily mean that the outcomes were attributed to chance, but rather to a successful investment of effort in an otherwise difficult task. Related to the argument we have raised above, mastery can also develop rather slowly or level out at a certain point without reaching perfection with people still being able to shape the odds of successes to their benefit. In our case, random guessing and thus capitalizing on chance when identifying the brightest square would lead to much lower probabilities in winning than 50%. People should experience mastering the task to a certain degree if they achieve these hit rates overall. This is also reflected in our ratings of control, which operationalize control beliefs close to the concept of internal control as defined in the literature (Ajzen, 2002; Rotter, 1966): Participants were explicitly asked to which degree they believed they could influence the outcome of the task (i.e. something one could relate to some degree of mastery) - if people would feel no mastery at all and think that outcomes were purely based on chance, they would also not think that they are able to influence the outcome.

Besides this, we very much appreciate your notion on the temporal dynamics in mastery to drive affective response and thought about how to also empirically examine the effects of temporal dynamics in mastery based on the success history within our study. However, our experimental paradigm was designed in a manner so that the different types of feedback are presented with rather constant frequency. For the fMRI analyses this is an important methodological constraint, as we aimed for stable frequencies of events across the whole session to increase power and avoid filtering out low-frequency events during pre-whitening of the BOLD time-series. We have thought and thoroughly discussed about how to implement such analyses in our experiment and tried several procedures to investigate temporal dynamics in mastery. Due to the experimental design constraints discussed above, we were however running into the problem that there were only very few task outcomes preceded by a longer and consistent history of WIN outcomes, which would be indicative of improvement over time in the HC condition. In fact, the outcome sequence was designed in a fashion that not more than 3 consecutive WIN or noWIN outcomes would be presented in any condition. We therefore concluded that the present data does not allow for performing reliable analyses of how the

performance history influences affect ratings. In order to test such effects reliably, the experimental task would have to be optimized in a way that drives the temporal dynamics of mastery, inducing “winning streaks” and “losing streaks”.

Comment #3: In the fMRI analyses, self-value mask removes value regions from the self ROIs. However, part of the argument being made is that the high control condition is subjectively valuable compared to the other conditions, and indeed the findings from study 3 show this. What is the rationale behind conducting the HC>MC analysis only in self regions rather than valuation (or a combination of both) given the subjective value involved in greater degree of control?

This is a very good point indeed and this analysis was in fact erroneously missing in the initial version of the submission. We have included the results of this analysis in the revised manuscript as follows.

Results:

II. 239-245: “Furthermore, we found significant activations in aspects of the vmPFC covered by the $\text{SELF} \cap \text{VALUE}$ mask (covering regions associated with both “self-referential” and “value”, -8,48,-12, $t(114)=6.13$, $Z=5.69$, $k=164$) as well as the $\text{VALUE} - \text{SELF}$ mask (-6,52,-12, $t(114)=5.71$, $Z=5.35$, $k=133$; all p-values $<.05$, small-volume FWE-corrected). These findings indicate, that, as hypothesized, cortical regions associated with processing of self-relevant information and value are engaged when outcomes are obtained in task environments that are perceived as controllable and thus provide information about the subjects’ capabilities (Northoff & Bermpohl, 2004; Tracy & Robins, 2007).”

Comment #4: The logic behind the conjunction of value of winning under high control is confusing. Why not simply run a conjunction of HC W > HC NoW and HC W > MC W, or an interaction between HC > MC and W > NoW? The prediction is that Winning under HC will generate larger responses in valuation and/or self-processing regions, which is a spreading interaction. The HC W > MC NoW contrast in the conjunction, to me, does not seem to add meaningful information in the conjunction, other than possibly help increase the effect in vMPFC.

Thank you for bringing this up and we apologize for not making our rationale any clearer. As we understand, your comment addresses two different questions: first, you question the validity of the specific configuration of the conjunction analysis we have conducted and second, you suggest that testing a spreading interaction of success and control would be also reasonable to do. We want to answer both aspects separately.

First, we want to point out that integrating HC:WIN > MC:noWIN in the conjunction analysis does not formally invalidate our results and does not help increasing the effect in the vmPFC. It is important to consider that the conjunction analyses we have applied is based on the minimum of several t-statistics. Including additional contrasts that show strong effects do not change the overall results of the conjunction as only the smallest effect defines the statistical significance of the analyses. As a result, the conjunction analysis with and without the HC:WIN > MC:noWIN produce the exact similar result. Nonetheless one might argue whether the composition of the conjunction is easy to follow and confusing. Here, we agree with the reviewer, that we have not made the rationale quite clear and were over inclusive in the setup of the conjunction. The more straightforward step would be to only include the contrasts

HC:WIN>MC:WIN and HC:WIN>HC:noWIN. Please apologize for the confusion and we have now updated the figure 5 and the description in the result section accordingly.

Your comment also aligns with comment #5a of reviewer #1 questioning the use of the conjunction analyses in general. Please also see our reply to this comment above for a more in-depth argument for why we believe that the conjunction analysis is appropriate if we aim for identifying converging additive effects in the BOLD response. In summary, this breaks down into the question whether to model *additive* or *super-additive effects* of success and control as you have already suggested. We believe the conjunction analyses address the more general hypotheses that control and success are jointly processed in brain regions rather than integrated in a super-additive manner or spreading activation, which is an even more specific assumption. Nonetheless, we think it is worth testing both models and we have now also conducted the test for the interaction and super-additive effects as you have suggested but did not find evidence for a spreading activation.

Results:

II. 246-261: “Next, we were interested in the additive effects of internal control and winning, by describing specific neural correlates of receiving positive outcomes under high control beliefs. To do so, we composed the two-way conjunction contrasting success achieved under high control to both success achieved under medium control and no success obtained under high control ($[HC:WIN>HC:noWIN] \cap [HC:WIN>MC:WIN]$; figure 5). This analysis revealed that the left vmPFC, specifically, is more strongly engaged when subjects’ successes are attributable to their own actions (-4,48,-10, $t(114)=3.45$; $Z=3.36$, $k=12$, $p=.022$, FWE-corrected within the $SELF \cap VALUE$ mask). The additivity of the effect in this region is illustrated by the common of responses to winning (WIN>noWIN: -2,48,-4, $t(114)=7.60$; $Z=6.82$, $k=193$; $p<.05$, FWE-corrected) as well as control (HC>MC; -8,48,-12, $t(114)=6.13$; $Z=5.69$, $k=162$, $p<.05$, small-volume FWE-corrected). The localization of this effect thus mirrors how the processing of value and self-reference converges in the vmPFC53,60. Aside from these rather additive effects of success and internal control beliefs, one might hypothesize that vmPFC activity responds in a super-additive fashion or spreading activation to the convergence of these factors. That is, vmPFC activity might differ more strongly between WIN and noWIN for HC than for MC outcomes. We thus tested this interaction ($[HC:WIN>HCnoWIN]>[MC:WIN>MCnoWIN]$), which did not yield any significant effects in any of the masks (even at more lenient thresholds at $p<.001$, uncorrected), nor on the whole-brain level.”

Methods:

II. 873-877: “For the analyses at the second level, we focused on the outcome phase of the task by implementing a within-subjects ANOVA including the individual first-level contrast images for MC:WIN, MC:noWIN, HC:WIN, and HC:noWIN. Using this approach we computed main effects of condition (HC>MC) and outcome valence (WIN>noWIN), the conjunction $(HC:WIN>HC:noWIN) \cap (HC:WIN>MC:WIN)$, as well as the interaction $((HC:WIN>HC:noWIN) > (MC:WIN>MC:noWIN))$.”

Figure 5:

Figure 5. Additive effect of winning under high control and parametric effects of emotion ratings. **a** Activations show the specific effect of HC:WIN (i.e. 2-way conjunction [HC:WIN>HC:noWIN] ∩ [HC:WIN>MC:WIN]) displayed at $p < .005$, uncorrected. The conjunction contrast survived small-volume FWE-correction at $p < .05$ inside the SELF∩VALUE mask in vmPFC. Bar plot represents differences of neural activation in vmPFC between HC:WIN and the other three outcomes. **b** Parametric modulation effect of happiness (red-yellow) and pride ratings (blue-green) at outcome presentation, displayed at $p < .001$, uncorrected for illustrative purposes. Bar plot shows group-level parameter estimates of the parametric modulations at the peak of each parametric effect inside the SELF∩VALUE mask. Violet outlines show SELF∩VALUE mask. Error bars are ± 1 standard error of the mean.

Table S19-S20:

Table S19. 2-way conjunction small-volume-corrected inside SELF∩VALUE, critical T = 3.164

set		cluster			peak					MNI coordinates (mm)			
p	c	p (FWE)	p (FDR)	k	p (unc)	p (FWE)	p (FDR)	T	Z	p (unc)	x	y	z
.050	1	.025	.486	12	.486	.022	.435	3.45	3.36	<.001	-4	48	-10

Note. Effects listed are for the contrast [HC:WIN>HC:noWIN] ∩ [HC:WIN>MC:WIN]

Table S20. 2-way conjunction inside SELF∩VALUE, $p < .005$, uncorrected

set		cluster			peak					MNI coordinates (mm)			
p	c	p (FWE)	p (FDR)	k	p (unc)	p (FWE)	p (FDR)	T	Z	p (unc)	x	y	z
1	2	.986	.911	80	.178	.984	.691	3.45	3.36	<.001	-4	48	-10
						1.000	.772	2.97	2.91	.002	-2	42	0
		1.000	.911	1	.911	1.000	.692	3.06	3.00	.001	0	42	4

Note. Effects listed are for the contrast [HC:WIN>HC:noWIN] ∩ [HC:WIN>MC:WIN]

Comment #5: In order to test whether vmPFC activity tracks pride more strongly than happiness ratings, why not conduct a contrast of pride > happiness, which is consistent with the rest of the analyses presented in the manuscript? The analysis in its current form compares beta weights for pride vs. happiness in the Self and Value ROIs. Alternatively, the effects of Winning and Control could also be tested using this approach, by extract beta weights from the ROIs and test the effects of Winning (W/NoW) and Control (HC/MC) rather than running contrasts. The differences in analytic approach raises questions about the robustness of the effect.

We agree with the reviewer that comparing the extracted beta-weights for the parametric modulations is not consistent with the remainder of the analyses. We have therefore performed the equivalent analysis again, testing the effect with the same approach as in the rest of the task effects. In fact, this analysis did not produce a significant difference between the parametric modulations of pride and happiness. We have updated the report of this analysis which now concludes that the parametric modulation effects are not significantly different from one another ($p = .189$, corrected). Please see also the updated figure 5 above. Also, we discuss these results in light of previous findings of vmPFC activity being associated with positive affect.

Results:

II. 279-284: “The effect for pride ratings survived FWE-correction on the whole-brain level (-2,56,0, $t(38)=6.15$, $Z=5.09$, $k=24$, $p=.008$), while the one for happiness did not. However, directly contrasting the parametric modulation effects of pride against happiness did not yield significant results in the vmPFC mask ($t(38)=4.83$, $Z=4.24$, $p=.189$, small-volume FWE-corrected inside the SELF \cap VALUE mask). This is very much in line with previous literature highlighting the involvement of vmPFC activity in the experience of positive affect (Wager et al., 2008), here demonstrated for pride and happiness.”

Discussion:

II. 421-426: “Here, the vmPFC seems to play a key role in tracking success, internal control beliefs as well as affective dynamics. In addition, activity in these regions is more strongly related to functional responses of brain regions processing reward information, i.e. ventral striatum, when outcomes depend on one’s own behavioral performance. In a final step, we highlight the subjective value of the emerging positive affect, as people discount monetary gains for tasks offering internal control in dependence of their pride response during task execution.”

II. 473-478: “Third, intra-individual variability in pride and happiness in response to task outcomes similarly covaried with neural activity in vmPFC, very much in line with previous ideas on how the vmPFC is involved in the affective valuation of outcomes in more or less controllable conditions (Kool et al., 2013). Collectively, these findings support the notion that positive affect and self-evaluation might depend on the integration of outcome information with self-representations, mediated via regions of the cortical midline and AG, mirroring findings of studies investigating other types of self-conscious affect (Müller-Pinzler et al., 2015).”

Comment #6: For PPI analyses, it wasn’t clear whether they tested for connectivity differences (e.g., between striatum and vmPFC) between conditions HC > MC, or whether additionally there were significant PPI effects within the HC condition itself. A significant difference in connectivity doesn’t address whether there is a significant connectivity effect in the HC condition itself, just that it is different across conditions.

Thank you for this comment which also aligns with the comment #5b & c of reviewer #1 and please apologize for not having described our PPI approach in sufficient detail. In the original submission, we have in fact only contrasted the connectivity of the VS between the HC and MC condition and have not checked for the connectivity within the HC condition itself. Following your recommendations, we have now tested the PPI effects in HC and MC separately to explore the significance of the functional connectivity within conditions. Both analyses support the interpretation that BOLD signal correlations of the VS with the vmPFC, CMS, and the TPJ are larger in HC and that these correlations are in fact statistically significant. Below you can find an illustration of the PPI effect for HC and MC clearly showing more widespread and significant effects in HC than in MC. We have added references to these additional control analyses in the result section and included them in supplementary tables S34 & S35, as well as supplementary figure S2 (see below). In addition, please see also the results of the various other control analyses we have conducted in response to comment #5c of reviewer #1 above.

Results:

II. 299-305: “In a set of control analyses we tested whether dynamics in functional connectivity were driven by WIN or noWIN outcomes. Comparing the PPIs contrasting

HC:WIN>MC:WIN and HC:noWIN>MC:noWIN did not reveal significant differences in the connectivity dynamics in any of the a-priori masks nor on the level of the whole brain. Further analyses additionally showed that the HC>MC PPI effects were driven by increased functional connectivity of left VS for HC outcomes (see supplementary tables S34 & S35, and supplementary figure S2).

Methods:

II. 912-929: “In a first follow-up analysis, we focused on whether the PPI described above was driven by WIN or noWIN outcomes. Therefore, we constructed two PPI terms for each subject by computing the interaction of the left VS time-course with the contrasts HC:WIN>MC:WIN and HC:noWIN>MC:noWIN, respectively. Aside from these two PPI terms, we included regressors modelling the two task phases, two rating phases, the two contrasts used for constructing the PPI terms, the activation time-course of the left VS, as well as six realignment parameters and their first derivatives. The first level images contrasting the WIN and noWIN PPI terms were then taken to a second-level one-sample t-test, to test whether there were significant connectivity differences for WIN and noWIN outcomes.

A second set of follow-up analyses tested for functional connectivity of the left VS only during HC and only during MC outcomes. For each subject, we computed the interaction of the left VS activation time-course with the contrast HC>0 and the contrast MC>0, respectively. We then set up two GLMs for each subject, including, aside from the respective PPI terms, regressors modelling the two task phases, two rating phases, the contrast used for constructing the PPI term, the activation time-course of the left VS, as well as six realignment parameters and their first derivatives. In addition, for the PPI modelling HC connectivity, the outcome phases for MC (MC:WIN, MC:noWIN) were included as two additional regressors. Vice versa, for the PPI modelling MC connectivity, the outcome phases for HC (HC:WIN, HC:noWIN) were included as two additional regressors. We then computed two second-level one-sample t-tests on the first level PPI effects (contrast images for HC PPI and MC PPI).”

Supplementary figure S2:

Figure S2. Functional connectivity of the left ventral striatum (VS). **a** For HC outcomes, the PPI term was based on the interaction of the left VS time-course and the contrast HC>0. Only positive connectivity effects were found. **b** For MC outcomes, the PPI term was based on the interaction of the left VS time-course and the contrast MC>0. Connectivity effects for this PPI were generally weak, so the effects are displayed on very liberal thresholds (p<.05, uncorrected) for illustration purposes. Blue-green indicates negative connectivity effects, red-yellow indicates positive connectivity effects.

Supplementary tables S34-S35:

Table S34. Left VS PPI (HC>0) wholebrain p<.001, uncorrected, k>=50

set		cluster				peak				MNI coordinates (mm)							
p	c	p(FWE)	p(FDR)	k	p(unc)	p(FWE)	p(FDR)	T	Z	p(unc)	x	y	z				
<.001	10	<.001	<.001	2973	<.001	.028	.276	5.75	4.83	<.001	-10	26	60				
						.033	.276	5.68	4.79	<.001	-8	56	38				
						.297	.499	4.76	4.18	<.001	4	38	26				
						.256	.499	4.83	4.23	<.001	-40	8	56				
						.532	.527	4.44	3.95	<.001	-42	16	52				
						.819	.648	4.09	3.69	<.001	-32	14	50				
						.262	.499	4.82	4.22	<.001	48	-64	42				
						.845	.675	4.06	3.66	<.001	52	-56	34				
						.970	.740	3.77	3.44	<.001	46	-60	30				
						.277	.499	4.79	4.20	<.001	42	28	-6				
						.379	.499	4.63	4.09	<.001	46	36	-12				
						.419	.499	4.58	4.05	<.001	34	22	-16				
						<.001	.001	572	<.001	.293	.499	4.76	4.18	<.001	-40	40	-6
										.385	.499	4.63	4.08	<.001	-38	34	2
										.567	.544	4.40	3.92	<.001	-44	42	-16
.124	.107	159	.015	.343	.499	4.69	4.13	<.001	-46	-68	40						
.002	.003	466	<.001	.444	.499	4.55	4.03	<.001	-4	-4	6						
				.473	.499	4.51	4.00	<.001	-10	14	6						
				.596	.544	4.37	3.90	<.001	-14	4	12						
.013	.015	310	.001	.481	.499	4.50	3.99	<.001	8	-56	34						
				.882	.691	3.99	3.62	<.001	0	-56	36						
				.920	.702	3.92	3.56	<.001	-4	-70	56						
.657	.636	56	.125	.683	.550	4.27	3.82	<.001	-28	20	-20						
.465	.413	79	.073	.914	.702	3.93	3.57	<.001	40	-60	-26						
				.976	.764	3.74	3.42	<.001	30	-66	-24						

Table S35. Left VS PPI (MC>0) wholebrain p<.001, uncorrected

set		cluster				peak				MNI coordinates (mm)			
p	c	p(FWE)	p(FDR)	k	p(unc)	p(FWE)	p(FDR)	T	Z	p(unc)	x	y	z
1	3	1.000	.858	1	.858	.992	.996	3.67	3.37	<.001	-20	-30	10
		.999	.858	2	.784	1.000	.996	3.37	3.13	<.001	-40	-86	14
		1.000	.858	1	.858	1.000	.996	3.33	3.09	<.001	-26	-88	18

Comment #7: In study 3, it makes sense that people should choose HC more often if the value is equal with MC, and that this should be related to pride. But the results also show that participants forgo monetary reward to choose HC (average value per trial for HC and MC, probably also in aggregate money forgone during the duration of the task). Shouldn't the degree to which an individual values control (and therefore gives up money to experience HC) relate to pride? This could be tested with the mean value of trials, money forgone during the task, or a point of subjective equality (the value at which people are equally likely to choose HC and MC, which will veer towards HC). It would be good to know if pride correlates with the bias towards HC over MC when their values are equal, and/or whether pride correlates with the propensity to forgo rewards to choose HC over MC.

Thank you for this comment, which we believe is a very good addition to our current set of analyses. We very much agree that we have previously provided a rather limited perspective on how to relate pride experiences with choice behavior and included only the arguably most straightforward parameter from the logistic regression models. Your suggestions are very helpful and we have now computed the different measures in order to relate the pride response to choice behavior.

In addition to our previous analyses, we have now also computed the point of equivalence (POE) indicating subjective equality as derived from the b0 and b1 parameters of the regression model and a second variable representing the monetary value of control reflecting the money forgone due to preferences for HC. The POE is the (log-transformed) monetary value offered for HC relative to that for MC, at which a given subject was equally likely to choose either option, i.e. had a probability of .5 to choose HC. Thus, we subtracted .5 from each subject's

logistic function and then used the *fzero* function in Matlab to determine the POE. Lower values indicate greater preference for HC and higher values indicate a preference for MC. The monetary value of control was based on the trial-by-trial difference between the presented offers for HC and MC, and the median of these differences was taken as an index of control preference for each subject.

In agreement with our initial analyses these variables also support that subjects prefer the HC condition. Participants had higher probability of choosing HC over MC if offers have equal monetary value, show equivalent probabilities for choosing either HC or MC only if MC offers exceed those of HC as indicated by the POE ($MD=-0.29$, $IQR=0.36$, Wilcoxon signed-rank $W=54$, one-sided $p<.001$), and are willing to give up money for expressing their preferences in the choices (monetary value of control: $MD=0.24€$, $IQR=0.20$, Wilcoxon signed-rank $W=1199$, one-sided $p<.001$; see supplementary table S5). Also, the pride response showed the expected association across the three variables with probability to choose HC in case of equal offers (Spearman's $\rho=.47$, one-sided $p<.001$), and the monetary value of control (Spearman's $\rho=.30$, one-sided $p=.026$) being significantly correlated even after the appropriate FDR correction for the handling of the three dependent variables. The association with POE was in the expected direction but did not reach significance after correcting for multiple comparisons (Spearman's $\rho=-.18$, one-sided FDR-corrected $p=.106$).

In addition to the more comprehensive description of the association between choice behavior and the pride response, we now also provide the correlations of individual differences in the experience of internal control and choice behavior. This was motivated by your and your co-referee's comments and our aim to strengthen our argument, replicate previous findings and better align our results to the existing literature on the subjective value of control, even though this was not explicitly addressed as a potential concern. Here, we correlated the three variables of choice behavior with differences in the perceived control between HC and MC and found significant associations comparable to those of the pride response in the sense that people who experience more difference in control between HC and MC also prefer HC over MC.

We have updated the results and methods section included your suggestions in the analyses.

Results:

II. 364-371: "Further exploration of variables indicative of the choice behavior showed similar evidence for a preference of HC over MC offers. First, the monetary value of control, i.e. the median difference in money offered for MC and HC on each trial (see methods) was significantly positive, indicating higher offers for MC than for HC (Wilcoxon signed-rank $W=1199$, one-sided $p<.001$, MD of difference= $0.23€$, $IQR=0.20$) and the willingness to forgo money to execute control. Second, the point of equivalence (POE), i.e. the (log-transformed) value of HC relative to MC at which subjects were equally likely to choose either option, was also significantly negative (Wilcoxon signed-rank $W=54$, one-sided $p<.001$, $MD=-0.29$, $IQR=0.36$; see supplementary table S5)."

II. 375-377: "Comparably, the POE also remained significantly negative when controlling for expected outcome probabilities (Wilcoxon signed-rank $W=283$, $p<.001$, $MD=-0.13$, $IQR=0.37$)."

II. 378-401: "A further goal of study 3 was to demonstrate that participants' preference for HC over MC varies in dependence of the affective dynamics during the main task. As expected, individuals who had a more positive pride response showed a higher probability to choose HC in the choice task, given equal values of the two choice options (Spearman's $\rho=.47$, one-sided $p<.001$; figure 7). Further, subjects with a

higher pride response also had a higher monetary value of control (Spearman's $\rho=.29$, one-sided $p=.029$) thus willing to forgo money in order to play the HC task. While the POE showed effects in the expected direction, it was not significantly correlated with the pride response after correcting for multiple comparisons (Spearman's $\rho=-.18$, one-sided $p=.106$; all p-values FDR-corrected). Notably, these associations cannot be fully explained by how difficult or exciting people experienced the HC and MC tasks, since after controlling for differences in these variables the overall pattern of association between task preference and pride response remained stable (see supplementary table S8). In order to further embed our data into the existing literature on perceived control and task preferences (Leotti et al., 2010; Wang & Delgado, 2019), we also tested the association between the three indices of control preference and individual differences in experienced control over the different tasks. Results clearly show that people who experienced stronger differences in control beliefs (HC-MC) also more strongly preferred the HC over the MC task. All three variables indicative of control preferences showed significant effects in the expected directions. Precisely, people experiencing stronger control were more likely to select the HC task in case of equal offers (Spearman's $\rho=.28$, one-sided $p=.048$), were willing to forgo money in order to play the HC task (Spearman's $\rho=.26$, one-sided $p=.048$), and showed shifts in the POE so that MC offers needed to be larger in order to achieve similar probabilities of selecting HC and MC (Spearman's $\rho=-.24$, one-sided $p=.048$, all p-values FDR-corrected for multiple comparisons). Collectively, these findings support and extend previous results that showed that mere choice is preferable to not having choices (Leotti & Delgado, 2011; Wang & Delgado, 2019) by highlighting the notion of instrumental control over the outcomes for building preferences (Ly et al., 2019), and emphasizing that self-related positive affect might be a relevant factor for mediating motivated behavior (Williams & DeSteno, 2008).

Methods:

II. 845-853: "In order to assess each subject's preference for control, we computed three indices for a preference of HC over MC. First, by setting β_1 to 0, β_0 directly defines the probability of choosing HC when monetary offers for HC and MC are equal ($p(\text{choose HC}|\text{HC=MC})$; see figure 7). Second, we determined the monetary value of control for each subject by computing the median of the trial-by-trial difference in money offered for choosing MC minus the offer for HC on that trial. Third, we computed the point of equivalence (POE), i.e. the (log-transformed) monetary value offered for HC relative to that for MC, at which a given subject was equally likely to choose either option, i.e. had a probability of .5 to choose HC. Thus, we subtracted .5 from each subject's logistic function and then used the `fzero` function in Matlab to determine the POE. In the present case, a more negative POE indicates a stronger preference for control."

Supplementary tables S5-S6:

Table S5. Pride responses, perceived control, and control preference (study 3)

	W	one-sided p	rank-biserial correlation	90% CI	
				lower	upper
Pride response ^a	1229.00	<.001	.93	.89	.96
Control difference HC-MC ^a	1252.00	<.001	.96	.94	.98
p(HC HC=MC) ^b	1185.00	<.001	.86	.77	.91
monetary value of control ^a	1199.00	<.001	.88	.80	.93
indifference point ^c	54.00	<.001	-.92	-.95	-.86

Note. a = alternative hypothesis specifies that the population median is larger than 0. b = Alternative hypothesis specifies that population median is larger than .5. c = Alternative hypothesis specifies that population median is smaller than 0.

Table S6. Associations between pride response, perceived control, and control preference

	Pride response					Control difference HC - MC				
	h.	rho	one-sided p*	90% CI		h.	rho	one-sided p*	90% CI	
				lower	upper				lower	upper
p(HC HC=MC)	+	.473	<.001	.267	.637	+	.282	.048	.050	.485
monetary value of control	+	.299	.026	.069	.500	+	.260	.048	.027	.467
indifference point	-	-.180	.106	-.398	.058	-	-.239	.048	-.449	-.003

Note. h.= hypothesized direction of correlation. *All p-values are FDR-corrected for multiple comparisons.

Minor points:

Comment #8: Related to my comment related to the relationship between pride and happiness, the abstract discusses the findings in relation to affective dynamics or positive affect, but the remainder of the results draw strong distinctions between pride and happiness. Why is there such a discrepancy in the description of the results across the manuscript?

Thank you very much for this comment and we agree with your evaluation on the discrepancies in how we refer to these concepts. In part, these inconsistencies relate to a previous comment of yours and your co-referee addressing the relation and distinction of the concepts of happiness and pride. Our motivation to include both happiness and pride as affective dimensions in our experiments was that we wanted to show that internal control believes result in a qualitative shift in affect when obtaining outcomes and are not merely related to stronger positive affect, in line with appraisal theories (Roseman et al., 1996; Scherer, 2001) and construal of self-conscious emotions (Lindquist & Barrett, 2008; Tracy & Robins, 2004). Happiness thus serves as a control or reference condition but as our result show and in line with theory it is also strongly affected by internal control believes. We have discussed this point more thoroughly above. As a result, we introduced inconsistencies with regards to how much we stressed the distinction of the affective constructs, made more prominent reference to their common core positive affect, or referred to the shift in the affective qualitative towards pride as a genuine self-evaluative affective response. We sincerely apologize for the resulting discrepancies. We have aimed to resolve this issue and thoroughly improved the manuscript to provide a more balanced description of the results also in line with the discussions we have had in the review process. You will find various changes in the manuscript related to this comment.

Comment #9: There is a broader literature on self and value (e.g., Somerville et al., JoCN, 2010; Hughes & Beer, JoCN, 2013; Korn et al., JNeuro 2012) that could further support the connection between vmPFC responses to self and reward.

We thank the referee for suggesting the additional literature on this topic. We have included references to the suggested (and additional, i.e. Chavez, Heatherton, & Wagner, 2017) manuscripts in the introduction in order to further strengthen our line of argument. Additionally, we included these studies in the discussion section, as they are helpful to understand how the vmPFC might subserve self-esteem regulation.

Discussion:

II. 519-524: “Furthermore, several studies have supported this role of vmPFC in the updating of self-related beliefs (Korn, Prehn, Park, Walter, & Heekeren, 2012; Kuzmanovic, Jefferson, & Vogeley, 2016; Kuzmanovic, Rigoux, & Tittgemeyer, 2018). For instance, Korn and coworkers (Korn et al., 2012) showed that a region in medial PFC was related to participants’ belief updating regarding their desirability as a person in response to social feedback. This collective evidence supports the interpretation that the vmPFC activation and affective dynamics we observed in response to outcomes from controllable tasks might reflect correlates of short-term updating of beliefs about one’s abilities.”

Comment #10: There is also an interesting experiment on reward responses associated with Illusory control that could be included given its relevance to the current study (Kool, Getz, Botvinick, JoCN, 2013).

This is a very interesting read indeed and we have integrated this literature in the discussion of illusory control.

Discussion:

II. 542-547: “Interestingly, previous studies have conceptually linked the vmPFC to illusions of control, distorted probability estimates, and positive affect, and there is evidence relating activity in this region to behavioral preferences for subjectively controllable tasks (Wang & Delgado, 2019). In concert with these findings, the present results provide a hint for clinical research to refocus on perturbed beliefs of control over the environment to understand affective and motivational symptoms as well as altered neural dynamics associated with relevant psychiatric conditions.”

Comment #11: The abstract introduces the term “self-evaluative affective dynamics” but this is very difficult to understand without a description.

We thank the reviewer for this remark. In line with your comment #8 we have now clarified our language and resolved discrepancies across the manuscript. Please find the revised abstract below.

“Experiencing events as controllable and attributing positive outcomes to own contributions is essential for human well-being. Based on classic psychological theory we test how internal control beliefs impact the affective valuation of task outcomes, neural dynamics and ensuing behavioral preferences. In three consecutive studies with

independent samples we show that affective dynamics increase, with a qualitative shift towards self-evaluative pride, when agents believe they caused a given task outcome. We demonstrate that these outcomes engage brain networks processing self-referential information in the cortical midline. Here, activity in the ventromedial prefrontal cortex tracks outcome valence regarding both success as well as internal control, and covaries with positive affect in response to outcomes. These affective dynamics also relate to increased functional coupling between the ventral striatum and cortical midline structures. Finally, we show that self-evaluative pride promotes preferences for control, even at a monetary cost. Our investigations extend recent models of positive affect and well-being, and emphasize that control beliefs drive intrinsic motivation.”

References

- Ajzen, I. (2002). Perceived Behavioral Control, Self-Efficacy, Locus of Control, and the Theory of Planned Behavior1. *Journal of Applied Social Psychology, 32*(4), 665–683. <https://doi.org/10.1111/j.1559-1816.2002.tb00236.x>
- Bandura, A. (1977). Toward a unifying theory of behavioral change. *Psychological Review, Vol. 84*, pp. 191–215. <https://doi.org/10.1037/0033-295X.84.2.191>
- Bartra, O., McGuire, J. T., & Kable, J. W. (2013). The valuation system: A coordinate-based meta-analysis of BOLD fMRI experiments examining neural correlates of subjective value. *NeuroImage, 76*(4), 412–427. <https://doi.org/10.1016/j.neuroimage.2013.02.063>
- Chavez, R. S., Heatherton, T. F., & Wagner, D. D. (2017). Neural Population Decoding Reveals the Intrinsic Positivity of the Self. *Cerebral Cortex, 27*(11), 5222–5229. <https://doi.org/10.1093/cercor/bhw302>
- Cockburn, J., Collins, A. G. E., & Frank, M. J. (2014). A Reinforcement Learning Mechanism Responsible for the Valuation of Free Choice. *Neuron, 83*(3), 551–557. <https://doi.org/10.1016/j.neuron.2014.06.035>
- Kable, J. W., & Glimcher, P. W. (2007). The neural correlates of subjective value during intertemporal choice. *Nature Neuroscience, 10*(12), 1625–1633. <https://doi.org/10.1038/nn2007>
- Knutson, B., Adams, C. M., Fong, G. W., & Hommer, D. (2001). *Anticipation of Increasing Monetary Reward Selectively Recruits Nucleus Accumbens. 21*, 1–5.
- Kool, W., Getz, S. J., & Botvinick, M. M. (2013). Neural Representation of Reward Probability: Evidence from the Illusion of Control. *Journal of Cognitive Neuroscience, 25*(6), 852–861. <https://doi.org/10.1162/jocn>
- Korn, C. W., Prehn, K., Park, S. Q., Walter, H., & Heekeren, H. R. (2012). Positively Biased Processing of Self-Relevant Social Feedback. *Journal of Neuroscience, 32*(47), 16832–16844. <https://doi.org/10.1523/JNEUROSCI.3016-12.2012>
- Kuzmanovic, B., Jefferson, A., & Vogeley, K. (2016). The role of the neural reward circuitry in self-referential optimistic belief updates. *NeuroImage, 133*, 151–162. <https://doi.org/10.1016/j.neuroimage.2016.02.014>
- Kuzmanovic, B., Rigoux, L., & Tittgemeyer, M. (2018). Influence of vmPFC on dmPFC Predicts Valence-Guided Belief Formation. *The Journal of Neuroscience, 0266–18*. <https://doi.org/10.1523/JNEUROSCI.0266-18.2018>
- Leotti, L. A., & Delgado, M. R. (2011). The Inherent Reward of Choice. *Psychological Science, 22*(10),

1310–1318. <https://doi.org/10.1177/0956797611417005>

- Leotti, L. A., & Delgado, M. R. (2014). The Value of Exercising Control Over Monetary Gains and Losses. *Psychological Science*, 25(2), 596–604. <https://doi.org/10.1177/0956797613514589>
- Leotti, L. A., Iyengar, S. S., & Ochsner, K. N. (2010). Born to choose: The origins and value of the need for control. *Trends in Cognitive Sciences*, 14(10), 457–463. <https://doi.org/10.1016/j.tics.2010.08.001>
- Lindquist, K. A., & Barrett, L. F. (2008). Constructing Emotion. *Psychological Science*, 19(9), 898–903. <https://doi.org/10.1111/j.1467-9280.2008.02174.x>
- Ly, V., Wang, K. S., Bhanji, J., & Delgado, M. R. (2019). A Reward-Based Framework of Perceived Control. *Frontiers in Neuroscience*, 13(February), 1–11. <https://doi.org/10.3389/fnins.2019.00065>
- Maier, S. F., & Seligman, M. E. P. (2016). Learned helplessness at fifty: Insights from neuroscience. *Psychological Review*, 123(4), 349–367. <https://doi.org/10.1037/rev0000033>
- Müller-Pinzler, L., Gazzola, V., Keysers, C., Sommer, J., Jansen, A., Frässle, S., ... Krach, S. (2015). Neural pathways of embarrassment and their modulation by social anxiety. *NeuroImage*, 119, 252–261. <https://doi.org/10.1016/j.neuroimage.2015.06.036>
- Müller-Pinzler, Laura, Czekalla, N., Mayer, A. V., Stolz, D. S., Gazzola, V., Keysers, C., ... Krach, S. (2019). Negativity-bias in forming beliefs about own abilities. *Scientific Reports*, 9(1), 1–15. <https://doi.org/10.1038/s41598-019-50821-w>
- Northoff, G., & Bermpohl, F. (2004). Cortical midline structures and the self. *Trends in Cognitive Sciences*, 8(3), 102–107. <https://doi.org/10.1016/j.tics.2004.01.004>
- O'Doherty, J. P., Buchanan, T. W., Seymour, B., & Dolan, R. J. (2006). Predictive Neural Coding of Reward Preference Involves Dissociable Responses in Human Ventral Midbrain and Ventral Striatum. *Neuron*, 49(1), 157–166. <https://doi.org/10.1016/j.neuron.2005.11.014>
- Pagnoni, G., Zink, C. F., Montague, P. R., & Berns, G. S. (2002). Activity in human ventral striatum locked to errors of reward prediction. *Nature Neuroscience*, 5(2), 97–98. <https://doi.org/10.1038/nn802>
- Roseman, I. J., Antoniou, A. A., & Jose, P. E. (1996). Appraisal determinants of emotions: Constructing a more accurate and comprehensive theory. In *Cognition and Emotion* (Vol. 10). <https://doi.org/10.1080/026999396380240>
- Rotter, J. B. (1966). Generalized expectancies for internal versus external control of reinforcement. *Psychological Monographs: General and Applied*, 80(1), 1–28. <https://doi.org/10.1037/h0092976>
- Roy, M., Shohamy, D., & Wager, T. D. (2012). Ventromedial prefrontal-subcortical systems and the generation of affective meaning. *Trends in Cognitive Sciences*, 16(3), 147–156. <https://doi.org/10.1016/j.tics.2012.01.005>
- Rutledge, R. B., Skandali, N., Dayan, P., & Dolan, R. J. (2014). A computational and neural model of momentary subjective well-being. *Proceedings of the National Academy of Sciences*, 1–6. <https://doi.org/10.1073/pnas.1407535111>
- Scherer, K. R. (2001). Appraisal Considered as a Process of Multilevel Sequential Checking. In K. R. Scherer, A. Schorr, & T. Johnstone (Eds.), *Appraisal Processes in Emotion: Theory, Methods, Research* (pp. 92–120). Oxford University Press.
- Seth, A. K. (2013). Interoceptive inference, emotion, and the embodied self. *Trends in Cognitive Sciences*, 17(11), 565–573. <https://doi.org/10.1016/j.tics.2013.09.007>
- Suzuki, S. (1997). Effects of number of alternatives on choice in humans. *Behavioural Processes*, 39(2), 205–214. [https://doi.org/10.1016/S0376-6357\(96\)00049-6](https://doi.org/10.1016/S0376-6357(96)00049-6)
- Tangney, J., Stuewig, J., & Mashek, D. (2007). Moral emotions and moral behaviour. *Annual Reviews of Psychology*, 58, 345–372. <https://doi.org/10.1146/annurev.psych.56.091103.070145.Moral>

- Tracy, J. L., & Robins, R. R. W. (2004). Putting the Self Into Self-Conscious Emotions: A Theoretical Model. *Psychological Inquiry*, 15(2), 103–125. https://doi.org/10.1207/s15327965pli1502_01
- Tracy, J. L., & Robins, R. W. (2007). The self in self-conscious emotions: A cognitive appraisal approach. *The Self-Conscious Emotions: Theory and Research*, (January), 3–20.
- Wager, T. D., Feldman Barrett, L., Bliss-Moreau, E., Lindquist, K. A., Duncan, S., Kober, H., ... Mize, J. (2008). The neuroimaging of emotion. In M. Lewis, J. M. Haviland-Jones, & L. Feldman Barrett (Eds.), *Handbook of emotions* (3rd ed., pp. 249–271). New York: Guilford Press.
- Wang, K. S., & Delgado, M. R. (2019). Corticostriatal Circuits Encode the Subjective Value of Perceived Control. *Cerebral Cortex*, 1–12. <https://doi.org/10.1093/cercor/bhz045>
- Williams, L. A., & DeSteno, D. (2008). Pride and perseverance: The motivational role of pride. *Journal of Personality and Social Psychology*, 94(6), 1007–1017. <https://doi.org/10.1037/0022-3514.94.6.1007>

Reviewers' Comments:

Reviewer #1:

Remarks to the Author:

The authors have clarified several points and added new analyses that enhance the paper. I appreciate the attention to details and the revisions the authors have implemented, the paper reads very well.

Reviewer #2:

Remarks to the Author:

I'd like to thank the authors for very thorough and clear set of responses to my comments. I feel that the manuscript was much improved by the revisions and additional analyses, and their responses made the contributions of the work clear. I'd be happy to cite this manuscript in my own work.

Sincerely,
Brent Hughes